# Magnesium efflux from Drosophila Kenyon cells is critical for normal and diet-enhanced long-term memory

Yanying Wu[1], Yosuke Funato[2], Eleonora Meschi[1], Kristijan D Jovanoski[1], Hiroaki Miki[2], Scott Waddell[1]*

[1]Centre for Neural Circuits and Behaviour, The University of Oxford, Tinsley Building, Oxford, United Kingdom; [2]Department of Cellular Regulation, Research Institute for Microbial Diseases, Osaka University, Suita, Japan

**Abstract** Dietary magnesium ($Mg^{2+}$) supplementation can enhance memory in young and aged rats. Memory-enhancing capacity was largely ascribed to increases in hippocampal synaptic density and elevated expression of the NR2B subunit of the NMDA-type glutamate receptor. Here we show that $Mg^{2+}$ feeding also enhances long-term memory in *Drosophila*. Normal and $Mg^{2+}$-enhanced fly memory appears independent of NMDA receptors in the mushroom body and instead requires expression of a conserved CNNM-type $Mg^{2+}$-efflux transporter encoded by the *unextended* (*uex*) gene. UEX contains a putative cyclic nucleotide-binding homology domain and its mutation separates a vital role for *uex* from a function in memory. Moreover, UEX localization in mushroom body Kenyon cells (KCs) is altered in memory-defective flies harboring mutations in cAMP-related genes. Functional imaging suggests that UEX-dependent efflux is required for slow rhythmic maintenance of KC $Mg^{2+}$. We propose that regulated neuronal $Mg^{2+}$ efflux is critical for normal and $Mg^{2+}$-enhanced memory.

*For correspondence:
scott.waddell@cncb.ox.ac.uk

**Competing interests:** The authors declare that no competing interests exist.

## Introduction

Magnesium ($Mg^{2+}$) plays a critical role in cellular metabolism and is considered to be an essential co-factor for more than 350 enzymes (*Romani and Scarpa, 2000*; *Vink and Nechifor, 2011*). As a result, alterations of $Mg^{2+}$ homeostasis are associated with a broad range of clinical conditions, including those affecting the nervous system, such as glaucoma (*DeToma et al., 2014*), Parkinson's disease (*Hermosura et al., 2005*; *Hermosura and Garruto, 2007*; *Lin et al., 2014*; *Shindo et al., 2016*), Alzheimer's disease (*Andrási et al., 2000*; *Andrási et al., 2005*; *Cilliler et al., 2007*; *Durlach et al., 1997*; *Glick, 1990*; *Lemke, 1995*; *Chui et al., 2011*; *Vural et al., 2010*), anxiety (*Sartori et al., 2012*), depression (*Whittle et al., 2011*; *Murck, 2002*; *Murck, 2013*; *Rasmussen et al., 1990*; *Ghafari et al., 2015*), and intellectual disability (*Arjona et al., 2014*).

Perhaps surprisingly, increasing brain $Mg^{2+}$ through diet can enhance neuronal plasticity and memory performance of young and aged rodents, measured in a variety of behavioral tasks (*Slutsky et al., 2010*; *Landfield and Morgan, 1984*; *Mickley et al., 2013*; *Abumaria et al., 2013*). In addition, elevated $Mg^{2+}$ reduced cognitive deficits in a mouse model of Alzheimer's disease (*Li et al., 2013*) and enhanced the extinction of fear memories (*Abumaria et al., 2011*). These apparently beneficial effects have led to the proposal that dietary $Mg^{2+}$ may have therapeutic value for patients with a variety of memory-related problems (*Billard, 2011*).

Despite the large number of potential sites of $Mg^{2+}$ action in the brain, the memory-enhancing property in rodents has largely been attributed to increases in hippocampal synaptic density and the activity of N-methyl-D-aspartate glutamate receptors (NMDARs). Extracellular $Mg^{2+}$ blocks the channel pore of the NMDAR and thereby inhibits the passage of other ions (*Mayer et al., 1984*;

**eLife digest** The proverbial saying 'you are what you eat' perfectly summarizes the concept that our diet can influence both our mental and physical health. We know that foods that are good for the heart, such as nuts, oily fish and berries, are also good for the brain. We know too that vitamins and minerals are essential for overall good health. But is there any evidence that increasing your intake of specific vitamins or minerals could help boost your brain power?

While it might sound almost too good to be true, there is some evidence that this is the case for at least one mineral, magnesium. Studies in rodents have shown that adding magnesium supplements to food improves how well the animals perform on memory tasks. Both young and old animals benefit from additional magnesium. Even elderly rodents with a condition similar to Alzheimer's disease show less memory loss when given magnesium supplements. But what about other species?

Wu et al. now show that magnesium supplements also boost memory performance in fruit flies. One group of flies was fed with standard cornmeal for several days, while the other group received cornmeal supplemented with magnesium. Both groups were then trained to associate an odor with a food reward. Flies that had received the extra magnesium showed better memory for the odor when tested 24 hours after training.

Wu et al. show that magnesium improves memory in the flies via a different mechanism to that reported previously for rodents. In rodents, magnesium increased levels of a receptor protein for a brain chemical called glutamate. In fruit flies, by contrast, the memory boost depended on a protein that transports magnesium out of neurons. Mutant flies that lacked this transporter showed memory impairments. Unlike normal flies, those without the transporter showed no memory improvement after eating magnesium-enriched food. The results suggest that the transporter may help adjust magnesium levels inside brain cells in response to neural activity.

Humans produce four variants of this magnesium transporter, each encoded by a different gene. One of these transporters has already been implicated in brain development. The findings of Wu et al. suggest that the transporters may also act in the adult brain to influence cognition. Further studies are needed to test whether targeting the magnesium transporter could ultimately hold promise for treating memory impairments.

*Bekkers and Stevens, 1993*; *Jahr and Stevens, 1990*; *Nowak et al., 1984*). Importantly, prior neuronal depolarization, driven by other transmitter receptors, is required to release the $Mg^{2+}$ block on the NMDAR and permit glutamate-gated $Ca^{2+}$ influx. The NMDAR therefore plays an important role in neuronal plasticity as a potential Hebbian coincidence detector. Acute elevation of extracellular $Mg^{2+}$ concentration ($[Mg^{2+}]_e$) within the physiological range (0.8–1.2 mM) can antagonize induction of NMDAR-dependent long-term potentiation (*Dunwiddie and Lynch, 1979*; *Malenka et al., 1992*; *Malenka and Nicoll, 1993*; *Slutsky et al., 2004*). In contrast, increasing $[Mg^{2+}]_e$ for several hours in neuronal cultures leads to enhancement of NMDAR mediated currents and facilitation of the expression of LTP (*Slutsky et al., 2004*). The enhancing effects of increased $[Mg^{2+}]_e$ were also observed in vivo in the brain of rats fed with $Mg^{2+}$-L-threonate (*Slutsky et al., 2010*). Hippocampal neuronal circuits undergo homeostatic plasticity (*Turrigiano, 2008*) to accommodate the increased $[Mg^{2+}]_e$ by upregulating expression of NR2B subunit containing NMDARs (*Slutsky et al., 2004*; *Slutsky et al., 2010*). The higher density of hippocampal synapses with NR2B containing NMDARs are believed to compensate for the chronic increase in $[Mg^{2+}]_e$ by enhancing NMDAR currents during burst firing. In support of this model, mice that are genetically engineered to overexpress NR2B exhibit enhanced hippocampal LTP and behavioral memory (*Tang et al., 1999*).

Olfactory memory in *Drosophila* involves a heterosynaptic mechanism driven by reinforcing dopaminergic neurons, which results in presynaptic depression of cholinergic connections between odor-activated mushroom body (MB) Kenyon cells (KCs) and downstream mushroom body output neurons (MBONs) (*Schwaerzel et al., 2003*; *Aso et al., 2010*; *Aso et al., 2012*; *Claridge-Chang et al., 2009*; *Burke et al., 2012*; *Liu et al., 2012*; *Plaçais et al., 2013*; *Owald et al., 2015*; *Hige et al., 2015*; *Barnstedt et al., 2016*; *Perisse et al., 2016*; *Aso et al., 2014*; *Owald and Waddell, 2015*). In addition, olfactory information is conveyed to KCs by cholinergic transmission from olfactory

projection neurons (*Yasuyama et al., 2002*; *Leiss et al., 2009*). Although it is conceivable that glutamate is delivered to the MB network via an as yet to be identified route, there is currently no obvious location for NMDAR-dependent plasticity in the known architecture of the cholinergic input or output layers (*Barnstedt et al., 2016*). The fly therefore provides a potential model to investigate other mechanisms through which dietary $Mg^{2+}$ might enhance memory.

The reinforcing effects of dopamine depend on the Dop1R D1-type dopamine receptor (*Kim et al., 2007*; *Qin et al., 2012*; *Handler et al., 2019*), which is positively coupled with cAMP production (*Tomchik and Davis, 2009*; *Boto et al., 2014*). Moreover, early studies in *Drosophila* identified the *dunce* and *rutabaga* encoded cAMP phosphodiesterase and type I $Ca^{2+}$-stimulated adenylate cyclase, respectively, to be essential for olfactory memory (*Dudai et al., 1976*; *Byers et al., 1981*; *Dudai and Zvi, 1984*; *Chen et al., 1986*; *Livingstone et al., 1984*; *Levin et al., 1992*). Studies in mammalian cells have shown that hormones or agents that increase cellular cAMP level often elicit a significant $Na^+$-dependent extrusion of $Mg^{2+}$ into the extracellular space (*Romani and Scarpa, 1990b*; *Romani and Scarpa, 1990a*; *Romani and Scarpa, 2000*; *Vink and Nechifor, 2011*; *Vormann and Günther, 1987*). However, it is unclear whether $Mg^{2+}$ extrusion plays any role in memory processing.

Here we demonstrate that *Drosophila* long-term memory (LTM) can be enhanced with dietary $Mg^{2+}$ supplementation. We find that the *unextended* (*uex*) (*Maeda, 1984*; *Coulthard et al., 2010*) gene, which encodes a functional fly ortholog of the mammalian Cyclin M2 $Mg^{2+}$-efflux transporter (CNNM) proteins, is critical for the memory enhancing property of $Mg^{2+}$. UEX function in MB KCs is required for LTM and functional restoration of *uex* reveals the MB to be the key site of $Mg^{2+}$-dependent memory enhancement. Chronically changing cAMP metabolism by introducing mutations in the *dnc* or *rut* genes alters the cellular localization of UEX. Moreover, mutating the conserved cyclic nucleotide-binding homology (CNBH) domain in UEX uncouples an essential role for *uex* from its function in memory. UEX-driven $Mg^{2+}$ efflux is required for slow rhythmic maintenance of KC $Mg^{2+}$ levels suggesting a potential role for $Mg^{2+}$ flux in memory processing.

## Results

### $Mg^{2+}$ feeding enhances LTM of wild-type flies

Prior studies reported that feeding rats with food containing a high concentration of $Mg^{2+}$-enhanced their learning and memory capability (*Slutsky et al., 2010*; *Landfield and Morgan, 1984*; *Abumaria et al., 2011*; *Mickley et al., 2013*; *Abumaria et al., 2013*). We therefore tested whether similar effects exist in flies by feeding them with food containing a high concentration of $Mg^{2+}$ before training. Surprisingly, wild-type flies fed for 4 days before training with food supplemented with additional magnesium chloride ($MgCl_2$) exhibited significantly enhanced 24 hr memory performance. Memory enhancement depends on concentration and was maximal when food was supplemented with 80 mM $MgCl_2$ (*Figure 1A*). Immediate memory performance was not obviously enhanced (*Figure 1B*). The enhancing effect of $MgCl_2$ was also observed in flies fed with magnesium sulfate ($MgSO_4$) but not calcium chloride ($CaCl_2$) (*Figure 1C*). In addition, feeding flies for 4 days with food containing between 5 and 80 mM strontium chloride ($SrCl_2$) resulted in high levels of mortality and flies that survived 5 mM $SrCl_2$ feeding did not show enhanced immediate or 24 hr memory performance (data not shown). The memory enhancing effects can therefore be specifically attributed to dietary supplementation of divalent $Mg^{2+}$.

### $Mg^{2+}$-enhanced memory is independent of NMDAR in the mushroom bodies

Since magnesium-L-threonate enhanced memory in rats was correlated with an upregulation of hippocampal NR2B subunit-containing NMDARs (*Slutsky et al., 2010*), we tested for changes in glutamate receptor expression in flies fed with $MgCl_2$. RT-qPCR analyses did not reveal a significant difference in the abundance of mRNAs for the putative NMDA (*Nmdar1*, *Nmdar2*), AMPA (*GluRIA*), or kainate-type (*GluRIIA*) receptors in heads taken from flies fed for 4 days with 80 mM $MgCl_2$ versus those fed with 1 mM $MgCl_2$ (*Figure 1D*).

We next directly tested whether $Mg^{2+}$-enhanced memory required NMDAR function, by knocking down expression of the *Nmdar1* or *Nmdar2* genes using transgenic UAS-driven RNA interference

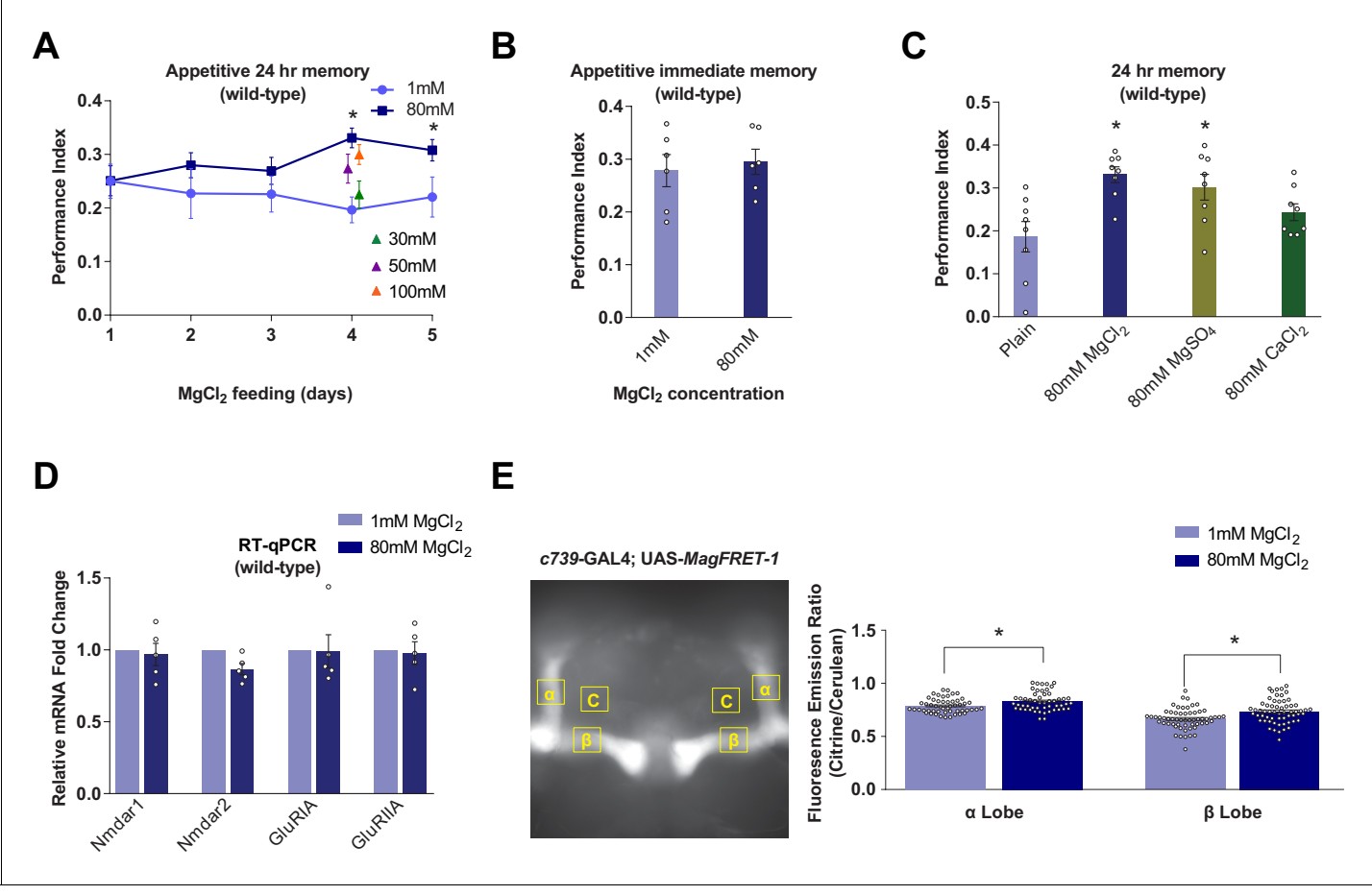

**Figure 1.** Dietary $Mg^{2+}$ supplementation enhances *Drosophila* long-term memory. (**A**) Wild-type flies were trained and tested for 24 hr appetitive memory after 1–5 days of ad libitum feeding on food supplemented with $Mg^{2+}$. Memory was significantly enhanced in flies fed for 4 days with 80 mM $MgCl_2$, as compared to those fed with 1 mM. 80 mM $MgCl_2$ produced marginally higher performance than 50 mM or 100 mM and so was considered optimal (asterisks denote $p < 0.05$, t-test between 1 mM and 80 mM groups for each time point, n = 6–8). (**B**) 4 days of 80 mM $MgCl_2$ food did not enhance immediate memory. (**C**) Appetitive 24 hr memory was enhanced by feeding wild-type flies for 4 days with $MgCl_2$ and $MgSO_4$, but not $CaCl_2$. Asterisks denote significant differences ($p < 0.05$, ANOVA, n = 6) between $Mg^{2+}$ fed and plain groups. (**D**) RT-qPCR showed no significant differences in glutamate receptor mRNA expression between 1 mM and 80 mM fed flies (t-test, n = 5). (**E**) *c739*-GAL4; UAS-*MagFRET-1* flies were fed for 4 days on food supplemented with $Mg^{2+}$. Brains were dissected and fixed and a fluorescence emission ratio measurement (Citrine/Cerulean) was taken as an indicator of $[Mg^{2+}]_i$. The MagFRET signal was significantly greater in the αβ lobes of flies fed with 80 mM $MgCl_2$ than those fed with 1 mM $MgCl_2$ ($p < 0.05$, t-test, n = 52–60). Unless otherwise noted, all data are mean ± standard error of mean (SEM). Asterisks denote significant differences ($p < 0.05$), individual data points displayed as open circles.

The online version of this article includes the following figure supplement(s) for figure 1:

**Figure supplement 1.** Knockdown of N-methyl-D-aspartate glutamate receptor (NMDAR) in mushroom bodies does not impair $Mg^{2+}$-enhanced memory.

(RNAi) constructs (*Dietzl et al., 2007*; *Perkins et al., 2015*). Of the two independent UAS-*Nmdar1*[RNAi] and four UAS-*Nmdar2*[RNAi] lines we tested, only one *Nmdar1*[RNAi] (BDSC 25941) line, when driven in all neurons by *neuronal Synaptobrevin* (*nSyb*)-GAL4, exhibited significantly decreased 24 hr memory performance, as compared to that of heterozygous control flies (*Figure 1—figure supplement 1A*). In contrast, more selective expression of this UAS-*Nmdar1*[RNAi] in LTM-relevant αβ KCs using *c739*-GAL4 did not significantly impair 24 hr memory performance (*Figure 1—figure supplement 1B*). Moreover, flies expressing *Nmdar1*[RNAi] in αβ neurons retained robust $Mg^{2+}$-enhanced memory (*Figure 1—figure supplement 1C*). These results suggest that $Mg^{2+}$-enhanced memory does not alter expression of glutamate receptors, or require NMDAR function in αβ KCs.

## Mg$^{2+}$ concentration in αβ neurons is elevated in flies fed high Mg$^{2+}$

We used MagFRET, the first genetically encoded fluorescent Mg$^{2+}$ sensor (*Lindenburg et al., 2013*), to test whether Mg$^{2+}$ feeding altered the intracellular Mg$^{2+}$ concentration ([Mg$^{2+}$]$_i$). We constructed flies harboring a UAS-*MagFRET-1* transgene and combined it with c739-GAL4 to express MagFRET-1 in αβ KCs. We compared the FRET signals in fixed brains from c739; UAS-*MagFRET-1* flies fed with either 1 mM or 80 mM MgCl$_2$ food for 4 days. The MagFRET signal was significantly higher in both the α and β collaterals of αβ KCs of flies fed with 80 mM, than in those fed with 1 mM (*Figure 1E*). This result indicates that Mg feeding elevates neuronal [Mg$^{2+}$]$_i$. Given the affinity of MagFRET-1 (Kd = 148 μM) and the ~50% increase in FRET signal upon Mg$^{2+}$ binding (*Lindenburg et al., 2013*), we estimate that the ~8% enhancement of the MagFRET signal measured in flies fed 80 mM MgCl$_2$ corresponds approximately to a 50 μM increase of αβ KC [Mg$^{2+}$]$_i$ on average.

## The *unextended* encoded CNNM-type Mg$^{2+}$ transporter has a role in memory

We identified *unextended* (*uex; Maeda, 1984*; *Coulthard et al., 2010*) as a gene altering appetitive olfactory LTM, reinforced with sucrose reward. Flies with the *uex*$^{MI01943}$ MiMIC insertion (*Venken et al., 2011*) showed a strong defect in 24 hr memory, but their performance immediately after training was indistinguishable from that of wild-type controls. More detailed analysis of *uex*$^{MI01943}$ flies revealed a steady decay of memory that first became significantly different to that of wild-type flies 12 hr after training (*Figure 2A*). No memory defect was evident in heterozygous *uex*$^{MI01943}$/+ flies, demonstrating that this putative *uex* allele is recessive.

*uex* piqued our attention because it is the single fly ortholog of the four human *CNNM* genes that encode Mg$^{2+}$ transporters (*Ishii et al., 2016*), and it also contains a putative CNBH domain that is structurally related to those in cyclic nucleotide-gated channels (*Zagotta et al., 2003*; *Flynn et al., 2007*; *Kesters et al., 2015*). Alignment of the 834 amino acid UEX sequence with CNNM1-4 reveals particularly high sequence conservation with CNNM2 and CNNM4 in the DUF21, CBS pair, and CNBH domains (*Figure 2—figure supplement 1A–C*). We therefore hypothesized that UEX had potential to link the memory-enhancing effects of dietary Mg$^{2+}$ with cAMP-dependent neuronal plasticity.

Although *uex*$^{MI01943}$ is assigned to the *uex* gene, the MiMIC element is annotated to lie 17 kb downstream of the *uex* coding region (*Venken et al., 2011*; *Figure 2B*). *RYa* (*Yoon et al., 2016*) is the next nearest gene to *uex*$^{MI01943}$ but is >230 kb further away. We first confirmed the MiMIC location by inverse PCR (*Attrill et al., 2016*). Importantly, no additional MiMIC insertion was detected in these flies. We next tested whether *uex*$^{MI01943}$ was responsible for the memory defect by precisely removing the MiMIC element by Minos transposase-mediated excision (*Arcà et al., 1997*; *Figure 2—figure supplement 2A and B*). MiMIC removal in *uex*$^{MI01943.ex1}$ and *uex*$^{MI01943.ex2}$ flies restored normal 24 hr memory performance, demonstrating that the MiMIC insertion is required for the *uex*$^{MI01943}$ memory defect (*Figure 2C*).

Both qRT-PCR of mRNA and western blot analysis of protein extracts from fly heads failed to reveal a significant difference in *uex*/UEX expression in *uex*$^{MI01943}$ flies. We therefore used CRISPR to introduce a stop codon into the fifth coding exon of the *uex* locus (*Figure 2B* and *Figure 2—figure supplement 2C*). Flies homozygous for the resulting *uex*Δ mutation were not viable as adults, dying at the larval stage. In contrast, heterozygous *uex*$^{MI01943}$/*uex*Δ flies were viable, but their 24 hr appetitive memory was significantly impaired (*Figure 2D*). These data demonstrate that *uex* is an essential gene and that *uex*$^{MI01943}$ is a viable hypomorphic allele of *uex*.

We also tested the aversive memory performance of *uex*$^{MI01943}$ mutant flies. Homozygous *uex*$^{MI01943}$ flies exhibited immediate memory that was indistinguishable from that of heterozygous and wild-type controls (*Figure 2E*). However, their 24 hr memory, formed following either five trials of aversive spaced training (*Tully et al., 1994*; *Jacob and Waddell, 2020*), or one trial of fasting facilitated training (*Hirano et al., 2013*), was significantly impaired (*Figure 2E*). These experiments suggest that *uex*$^{MI01943}$ flies are more generally compromised in their ability to form LTM. Unless otherwise specified, all subsequent analyses of memory in this study use appetitive sugar-rewarded conditioning.

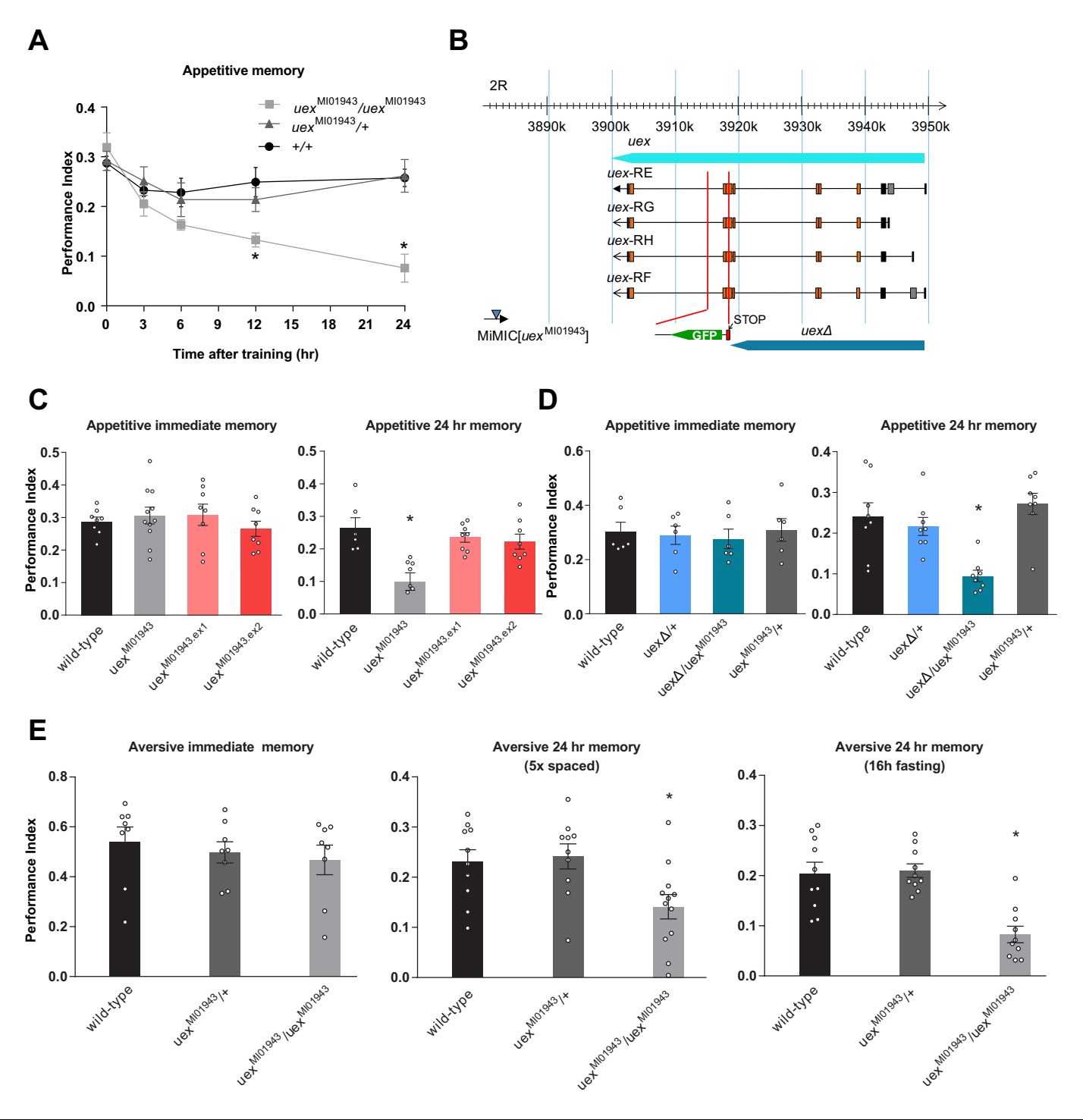

**Figure 2.** $uex^{MI01943}$ mutant flies have defective long-term memory (LTM). (**A**) Appetitive memory retention was tested at various times after training. Flies homozygous for $uex^{MI01943}$ showed a significant defect in memory from 12 hr after training, as compared to the performance of heterozygous $uex^{MI01943}$/+ and wild-type control flies (p<0.05, ANOVA, n = 6–10). (**B**) The $uex$ locus lies on chromosome 2R between 3,900,285 and 3,949,425 (light blue bar). The four alternate $uex$ transcripts, $uex$-RE, $uex$-RG, $uex$-RH, and $uex$-RF, all encode the same protein. The $uex^{MI01943}$ MiMIC (blue triangle) resides ~17 kb downstream of the $uex$ coding region. The CRISPR/Cas9 edited $uex\Delta$ allele replaces a 3047 bp fragment, including Exon 7 of $uex$ with a STOP signal (termination codon in all three reading frames) and a GFP cassette, truncating the $uex$ reading frame (dark blue bar). (**C**) Precise excision of the $uex^{MI01943}$ MiMIC restores normal 24 hr memory to $uex^{MI01943.ex1}$ and $uex^{MI01943.ex2}$ flies (p<0.05, ANOVA, n = 8–11). (**D**) $uex\Delta$ fails to complement the 24 hr memory defect of $uex^{MI01943}$ (p<0.05, ANOVA, n = 6–8). (**E**) Flies homozygous for $uex^{MI01943}$ showed a significant defect in aversive LTM, as

*Figure 2 continued on next page*

Figure 2 continued

compared to the performance of heterozygous *uex*[MI01943]/+ and wild-type control flies (p<0.05, ANOVA, n = 8–12). An LTM defect was also observed following five cycles of aversive spaced training and a 16 hr fasting facilitated one-cycle training protocol. Immediate aversive memory was unaffected in *uex*[MI01943] homozygous mutant flies.

The online version of this article includes the following source data and figure supplement(s) for figure 2:

**Source data 1.** Table of sugar and olfactory sensory acuity controls for all behavioral experiments in this manuscript.
**Figure supplement 1.** Conservation of UEX with its orthologs.
**Figure supplement 2.** Construction schemes for *uex Minos* excision and creation of the *uex*Δ allele.

## A role for *uex* in the mushroom bodies

To localize *uex* in the brain we first took advantage of VT23256-GAL4 transgenic flies, in which GAL4 is driven by an 853 bp sequence from the first intron of *uex* (*Kvon et al., 2014*). VT23256-driven UAS-*EGFP* revealed restricted expression in αβ KCs with particularly strong label in αβ core (αβ$_c$) neurons (*Figure 3A*). We also used CRISPR to insert a C-terminal HA-epitope tag into the *uex* open reading frame (*Figure 3—figure supplement 1A*). These flies were viable as homozygotes indicating that the resulting UEX::HA fusion protein retains function. Immunostaining flies harboring this *uex::HA* locus with an anti-HA antibody revealed prominent labeling of all the major KC classes in the MB, in addition to lower expression throughout the brain (*Figure 3B*). This *uex* expression profile is also supported by single-cell sequencing analyses (*Figure 3—figure supplement 1B*; *Croset et al., 2018*; *Davie et al., 2018*). Given the established role for αβ KCs in olfactory LTM (*Pascual and Préat, 2001*; *Yu et al., 2006*; *Krashes et al., 2007*; *Krashes and Waddell, 2008*), we reasoned that a mnemonic role for UEX may involve expression in KCs.

We next used GAL4-directed expression of RNAi to test whether 24 hr memory performance required *uex* in the MB. Flies expressing *uex*[RNAi] (*Perkins et al., 2015*) in all αβ KCs (c739-GAL4; *Yang et al., 1995*; *Perisse et al., 2013*) or only in αβ$_c$ KCs (NP7175-GAL4; *Tanaka et al., 2008*) showed normal immediate memory but significantly impaired 24 hr memory (*Figure 3C*). In contrast, *uex*[RNAi] expression in αβ surface (αβ$_s$, 0770-GAL4; *Perisse et al., 2013*) or α′β′ KCs (c305a-GAL4; *Krashes et al., 2007*) did not significantly alter immediate or LTM performance. Normal 24 hr appetitive memory performance is therefore particularly sensitive to *uex* expression in αβ$_c$ neurons. To reduce the likelihood that the *uex*[RNAi] associated memory defect results from a developmental consequence, we also restricted UAS-*uex*[RNAi] expression to adulthood using GAL80[ts]-mediated temporal control (*McGuire et al., 2003*). At permissive 18°C, GAL80[ts] binds to GAL4 and suppresses its transcriptional activator function. At restrictive 30°C, GAL80[ts] can no longer bind to GAL4, which frees GAL4 to direct expression of the UAS-*uex*[RNAi] transgene. Flies were raised through development at 18°C and moved to 30°C after eclosion. Restricting UAS-*uex*[RNAi] expression to αβ KCs in adult flies using c739-GAL4 with GAL80[ts] produced a similar 24 hr specific memory defect to that observed when UAS-*uex*[RNAi] was expressed without temporal control (*Figure 3D–F*). We assessed the efficacy of the UAS-*uex*[RNAi] knockdown using our tagged *uex::HA* locus. Brains from heterozygous *uex::HA* flies expressing *uex*[RNAi] in the αβ and γ KCs with MB247-GAL4 (*Zars et al., 2000*) were immunostained using anti-HA antibody. Comparing the intensity of immunolabeling in brains from *uex::HA*; MB247-GAL4/*uex*[RNAi] flies with that from *uex::HA*; MB247-GAL4/+ flies showed that *uex*[RNAi] expression significantly reduced anti-HA signal in the αβ and γ KCs (*Figure 3G and H*). This result demonstrates the efficiency of the *uex*[RNAi] transgene and the utility of the CRISPR/Cas9 edited *uex::HA* locus.

We next tested whether expression in specific KCs of an UAS-*uex* transgene could restore 24 hr memory capacity to *uex*[MI01943] flies. Memory performance of *uex*[MI01943] flies expressing UAS-*uex* in αβ and γ KCs (MB247-GAL4; *Zars et al., 2000*) or only the αβ KCs (c739-GAL4) was significantly improved over that of *uex*[MI01943] flies, and was statistically indistinguishable from that of controls with an intact *uex* locus (*Figure 4A*). In contrast, UAS-*uex* expression in α′β′, αβ$_c$, or αβ$_s$ KCs did not restore memory performance to *uex*[MI01943] flies and overexpressing *uex* in αβ KCs of wild-type flies did not augment 24 hr memory (*Figure 4A and B*). Normal 24 hr memory performance could also be restored to *uex*[MI01943] flies if UAS-*uex* expression was confined to c739-GAL4 neurons (all αβ KCs) in adulthood using GAL80[ts]-mediated temporal control (*Figure 4C and D*). Together, these loss-of-function RNAi and restoration experiments establish that UEX plays an important role in adult

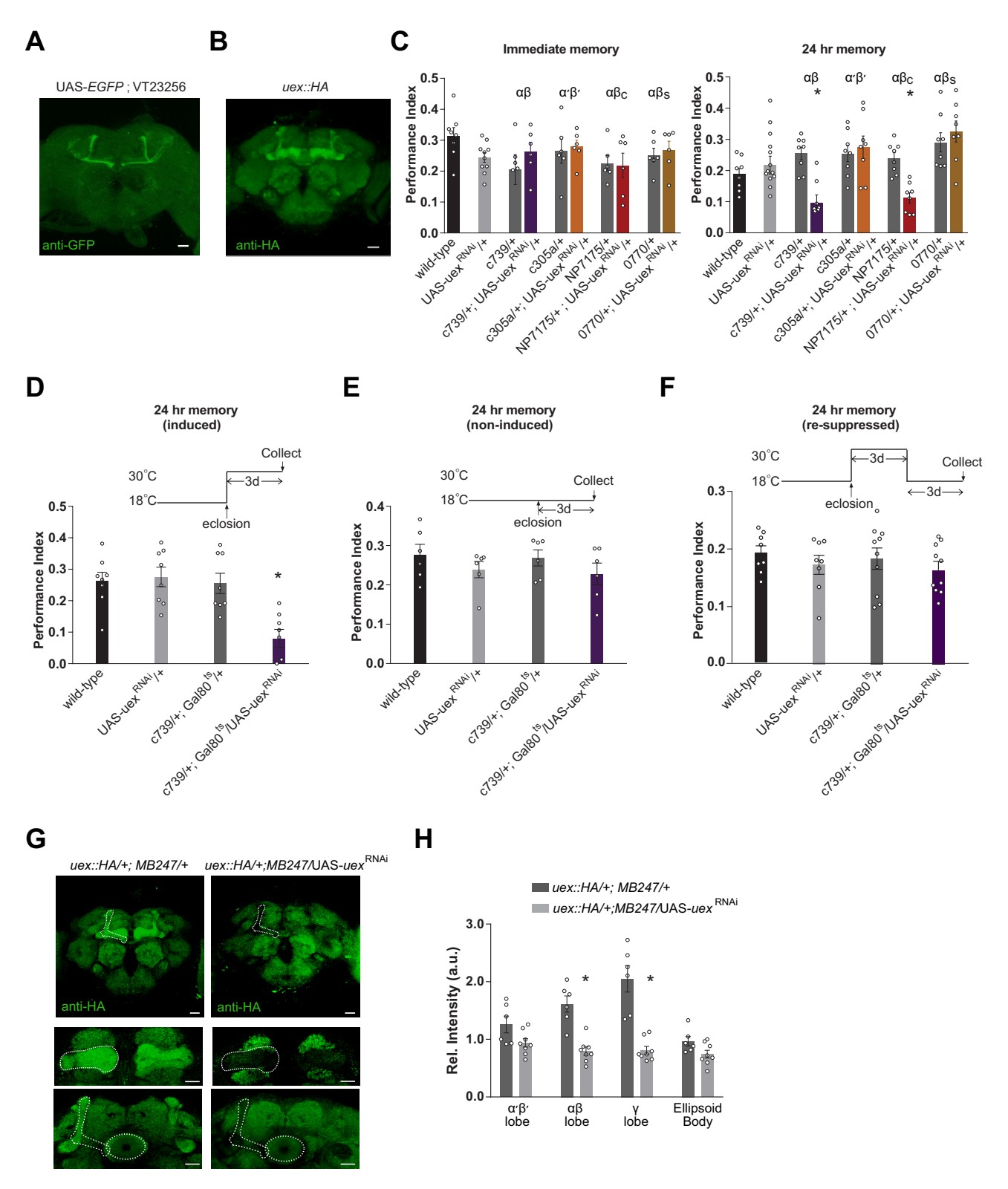

**Figure 3.** Knocking down *uex* expression in αβ Kenyon cells (KCs) impairs LTM. (**A**) A *uex* promoter fragment-GAL4 directs GFP expression in αβ_c KCs. Anti-GFP immunostained *uex*-GAL4 (VT23256); UAS-*EGFP* line. (**B**) Anti-HA immunostaining of brains harboring the CRISPR/Cas9-edited *uex::HA* locus shows strong labeling of UEX in all the major subdivisions of the mushroom body (MB). Scale bars 20 μm. (**C**) RNAi knockdown of *uex* in all αβ (*c739*-GAL4) or just αβ_c (*NP7175*-GAL4) KCs specifically impaired 24 hr memory. αβ_s (*0770*-GAL4) or α′β′ (*c305a*-GAL4) KC expression had no effect (p<0.05,
*Figure 3 continued on next page*

Figure 3 continued

ANOVA, n = 6–10 for immediate and n = 8–14 for 24 hr memory). (D) Defective LTM was observed if $uex^{RNAi}$ expression was confined to αβ KCs of adult flies using GAL80$^{ts}$-mediated temporal control. (E) LTM performance was unaffected if the $uex^{RNAi}$ was kept suppressed throughout and (F) LTM performance was restored to normal levels if expression of $uex^{RNAi}$ was re-suppressed for 3 days (p<0.05, ANOVA, n = 6 for immediate and n = 8 for 24 hr memory). (G) Immunostaining shows the effectiveness of $uex^{RNAi}$. Fluorescence intensity in the αβ and γ lobes of $uex::HA$ flies decreased significantly when UAS-$uex^{RNAi}$ was expressed with MB247-GAL4. Scale bars 20 μm. (H) Quantification of fluorescence intensity in G (p<0.05, t-test, n = 6–8). The online version of this article includes the following figure supplement(s) for figure 3:

Figure supplement 1. Construction scheme for the $uex::HA$ line and tSNE plots of $uex$ expression.

αβ KCs. Finding that $αβ_c$ RNAi knockdown of $uex$ produces a memory defect (*Figure 3C*) but UAS-$uex$ expression in $αβ_c$ does not rescue the $uex^{MI01943}$ mutant defect (*Figure 4A*) suggests that UEX function in $αβ_c$ KCs is essential for appetitive LTM, whereas both the $αβ_c$ and $αβ_s$ KCs need to have functional UEX to support LTM. In addition, the ability of UAS-$uex$ to restore performance to $uex^{MI0194}$ flies provides further support that $uex$ is responsible for the memory impairment in $uex^{MI01943}$ flies.

## $uex$ expression in the MB supports Mg$^{2+}$-enhanced memory

We next investigated whether Mg$^{2+}$ feeding (4 days with 80 mM MgCl$_2$) could improve memory performance in flies with compromised $uex$ function. Flies carrying the $uex^{MI01943}$ allele (*Figure 4F*) or those expressing UAS-$uex^{RNAi}$ in the αβ KCs with c739-GAL4 (*Figure 4E*) did not show enhanced memory when fed with 80 mM MgCl$_2$, as compared to flies fed with 1 mM MgCl$_2$. Moreover, the Mg$^{2+}$-enhanced memory was recovered in $uex^{MI01943}$ mutant flies when $uex$ expression was restored to the αβ KCs (*Figure 4F*). All control flies (c739-GAL4, UAS-$uex^{RNAi}$, and UAS-$uex$) with unperturbed $uex$ expression exhibited significantly enhanced memory when fed with 80 mM as compared to 1 mM MgCl$_2$. Overexpressing UAS-$uex$ in αβ KCs with c739-GAL4 in flies with a wild-type genetic background neither enhanced regular 24 hr memory (*Figure 4B*), or that in flies fed for 4 days with 40 or 80 mM MgCl$_2$ (*Figure 4G*). We also tested whether 4 days of 80 mM MgCl$_2$ supplementation enhanced 24 hr memory performance following aversive spaced training. Again, memory of wild-type, but not $uex^{MI01943}$ mutant flies showed enhancement (*Figure 4—figure supplement 1*). Together these results indicate that optimal memory enhancement with Mg$^{2+}$ feeding requires, and can be fully supported by, UEX function in αβ KCs.

## UEX is a functionally conserved magnesium transporter

Given the strong sequence conservation of UEX with mammalian CNNM2/4 we tested whether CNNM2 could functionally substitute for UEX and restore the LTM defect of $uex^{MI01943}$ flies. Several point mutations in CNNM2 have been identified in human patients with hypomagnesemia, which is associated with brain malformation and intellectual disability (*Arjona et al., 2014*). Introduction of the equivalent mutations into mouse CNNM2 (CNNM2$^{E357K}$, CNNM2$^{T568I}$, CNNM2$^{S269W}$, and CNNM2$^{E122K}$) showed that these patient-derived lesions impair magnesium transport (*Arjona et al., 2014*). We constructed flies carrying wild-type and these mutant variant UAS-CNNM2 transgenes (*Figure 5A*). Staining for an associated C-terminal HA-tag revealed clear expression of all UAS-CNNM2::HA variants in αβ neurons when driven with c739-GAL4 (*Figure 5—figure supplement 1*). However, only expression of wild-type CNNM2, and not point-mutant forms, in αβ KCs of $uex^{MI01943}$ mutant flies restored 24 hr memory performance (*Figure 5B*).

We also tested whether UEX can mediate Mg$^{2+}$ extrusion. UEX expressed in HEK293 cells localized to the plasma membrane and cells loaded with Mg$^{2+}$ and the Mg$^{2+}$ indicator Magnesium Green showed rapid Mg$^{2+}$ efflux (*Figure 5—figure supplement 2* and *Video 1*), as compared to cells transfected with empty vector. Mg$^{2+}$ extrusion driven by UEX was noticeably less efficient than in cells expressing Human CNNM4 (*Figure 5—figure supplement 2*), which is known to have similar efficiency to CNNM2 (*Hirata et al., 2014*). However, we do not know if UEX and CNNM4 expression is equivalent. Nevertheless, demonstration of cross-species complementation and Mg$^{2+}$ efflux activity defines UEX as a functional homolog of mammalian CNNM2/4.

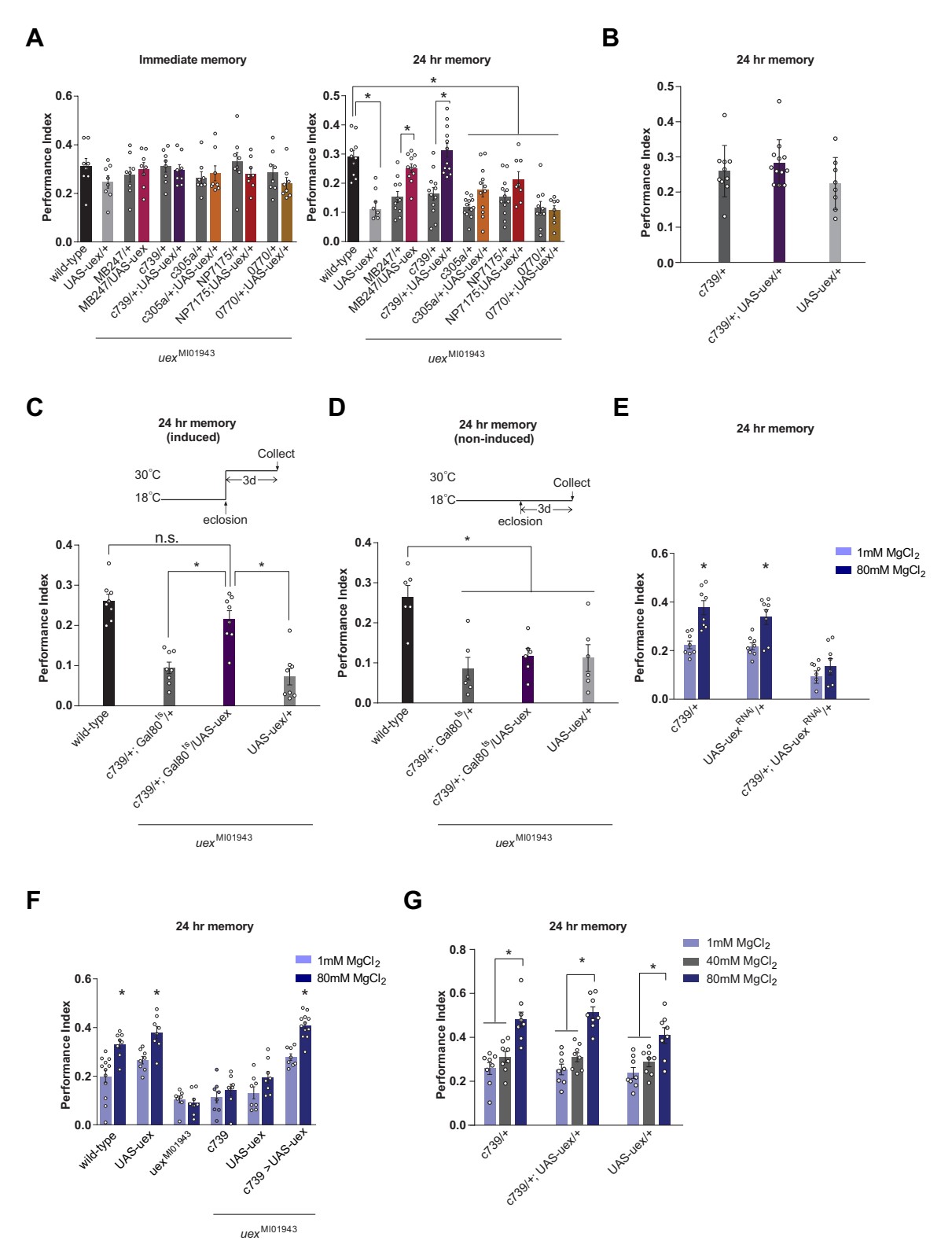

**Figure 4.** Rescue of the LTM defect in *uex*[MI01943] flies. Restoring expression of UAS-*uex* in αβ and γ (MB247-GAL4) or αβ Kenyon cells (KCs) rescued 24 hr memory performance of *uex*[MI01943] flies, whereas expression in αβ_c, αβ_s or α'β' KCs did not (p<0.05, ANOVA and t-test, n = 8–12). (**B**) Overexpression of UAS-*uex* in αβ KCs did not enhance 24 hr memory performance in wild-type flies (ANOVA, n = 8–12). (**C**) Defective LTM was rescued if UAS-*uex* expression was confined to αβ KCs of adult flies using GAL80[ts] mediated temporal control (p<0.05, ANOVA, n = 6 for immediate and n = 8

*Figure 4 continued on next page*

*Figure 4 continued*

for 24 hr memory) but (D) remained defective if UAS-*uex* expression was not released. (E) Memory enhancement with dietary Mg$^{2+}$ is supported by UEX in αβ KCs. Memory of flies expressing UAS-*uex*$^{RNAi}$ in the αβ KCs cannot be enhanced with Mg$^{2+}$ feeding (t-test, n = 8). (F) Memory of *uex*$^{MI01943}$ mutant flies cannot be enhanced with Mg$^{2+}$ feeding, but enhancement was restored by expressing UAS-*uex* in αβ KCs ($p < 0.05$, t-test, n = 8–12). (G) Memory of wild-type flies was not sensitized to Mg$^{2+}$ enhancement by overexpressing UAS-*uex* in αβ KCs. Memory was enhanced if the flies were fed with 80 mM MgCl$_2$, but not with suboptimal 40 mM MgCl$_2$ ($p < 0.05$, ANOVA, n = 8).

The online version of this article includes the following figure supplement(s) for figure 4:

**Figure supplement 1.** Mg$^{2+}$ feeding enhanced LTM after aversive spaced training in wild-type but not *uex*$^{MI01943}$ mutant flies.

## An intact CNBH domain is required for memory

Given the established role for cAMP signaling in memory-relevant plasticity in invertebrates and mammals (*Kandel, 2012*), we tested the importance of the CNBH domain in UEX. We constructed flies carrying a point-mutated CNBH UAS-*uex*$^{R622K}$ transgene (*Figure 6A*). The equivalent R622K amino acid substitution abolishes cAMP binding in the regulatory subunit of cAMP-dependent protein kinase, PKA (*Bubis et al., 1988*). Expressing UAS-*uex*$^{R622K}$ in αβ neurons with c739-GAL4 did not restore 24 hr memory performance, or alter the immediate memory performance, of *uex*$^{MI01943}$ mutant flies (*Figure 6B*).

We also used CRISPR to attempt to introduce the R622K mutation into the CNBH of the native *uex* locus (*Bassett et al., 2013*; *Gratz et al., 2013*; *Yu et al., 2013*). Unexpectedly, this approach did not introduce the R622K substitution but instead replaced T626 in the CNBH with NRR. Fortuitously, flies homozygous for this *uex*$^{T626NRR}$ allele were viable as adults, unlike those homozygous for *uex*Δ, suggesting that the *uex*$^{T626NRR}$ encoded UEX retains function. However, flies homozygous for *uex*$^{T626NRR}$ or heterozygous *uex*$^{T626NRR}$/ *uex*$^{MI01943}$ flies exhibited a strong 24 hr memory defect (*Figure 6C*). Immediate memory was also impaired in homozygous *uex*$^{T626NRR}$ flies, unlike flies carrying all other combinations of *uex* alleles. In addition, memory of *uex*$^{T626NRR}$ flies could not be enhanced with Mg$^{2+}$ feeding (*Figure 6D*). The *uex*$^{T626NRR}$ mutation therefore uncouples the essential

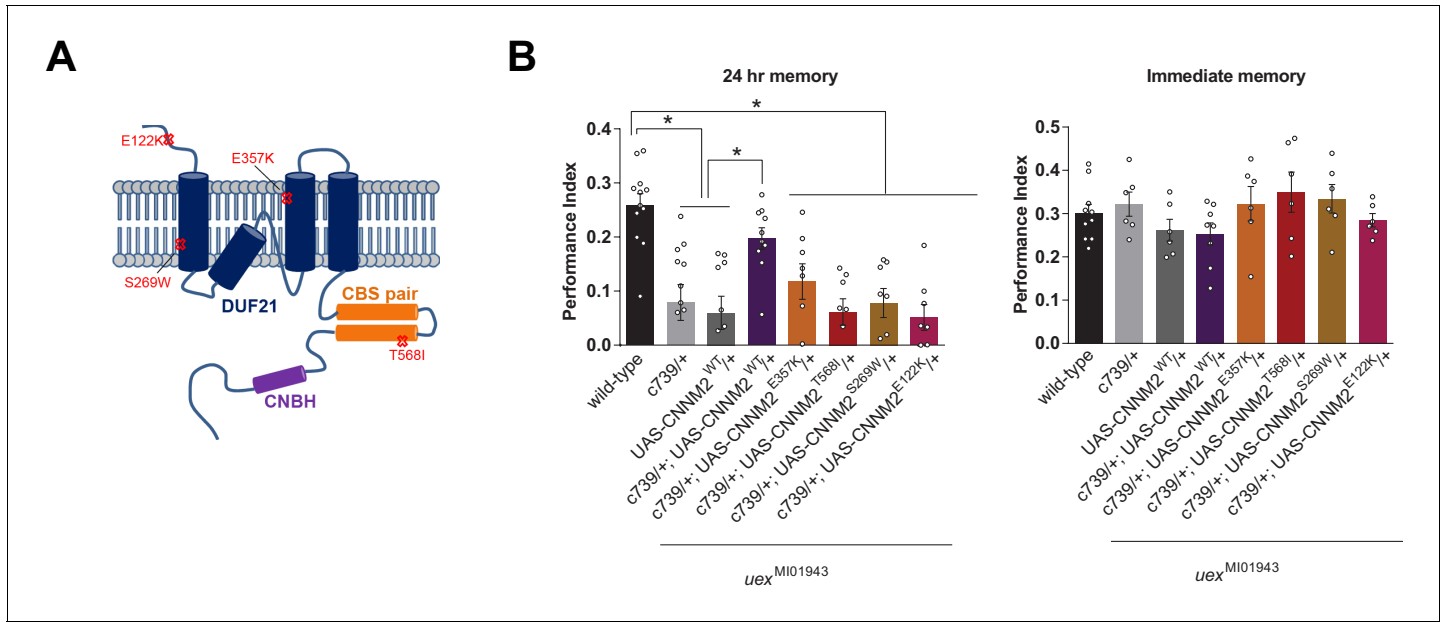

**Figure 5.** *uex* encodes an evolutionarily conserved Mg$^{2+}$ transporter. (A) Model of CNNM2 protein structure showing clinically relevant point mutations. Adapted and modified from *Arjona et al., 2014*. (B) Overexpression of wild-type, but not mutant, CNNM2 in αβ Kenyon cells rescues the memory defect of *uex*$^{MI01943}$ mutant flies ($p < 0.05$, ANOVA, n = 6–8 for immediate and n = 8–12 for 24 hr memory).

The online version of this article includes the following figure supplement(s) for figure 5:

**Figure supplement 1.** Transgenic expression of mutant variants of CNNM2.

**Figure supplement 2.** UEX-dependent Mg$^{2+}$ efflux in HEK293 cells.

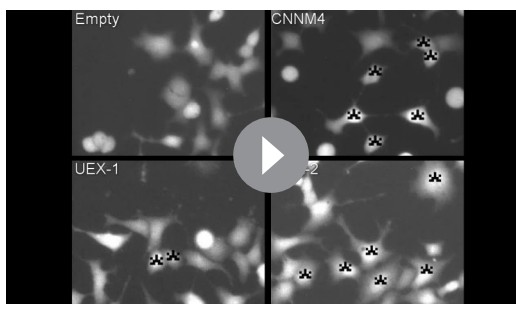

**Video 1.** UEX promotes $Mg^{2+}$-efflux from HEK293 cells. Representative movies showing $Mg^{2+}$-efflux from HEK293 cells transfected with different expression vectors. Imaging protocol is described in *Yamazaki et al., 2013*. The cells indicated with asterisks in the first frame of each movie are the cells expressing the anti-FLAG immunostained CNNM4 or UEX, which were identified after each live-imaging experiment. Empty vector control is shown in the upper left. The fluorescence signal of CNNM4-FLAG and UEX-FLAG expressing cells decreases rapidly when extracellular $Mg^{2+}$ is depleted, which was performed between the third and fourth frames in each movie.
https://elifesciences.org/articles/61339#video1

role for *uex* from a function in memory and suggests that cyclic nucleotide regulated activity is critical for UEX to support normal and $Mg^{2+}$-enhanced memory. Although we confirmed using western blotting that a full-length protein is expressed in $uex^{T622NRR}$ flies (*Figure 6E*), our antibody did not permit us to verify that the $UEX^{T626NRR}$ protein localizes appropriately in the brain. Further work is therefore required to characterize the cellular localization, cAMP binding, and $Mg^{2+}$ transport function of the protein encoded by this serendipitous $uex^{T626NRR}$ allele.

## Chronic cAMP manipulation alters UEX localization in KCs

We tested whether cAMP could acutely alter UEX activity by applying forskolin to UEX-expressing HEK293 cells. However, no obvious change in the UEX-dependent $Mg^{2+}$ efflux dynamic was observed (data not shown). We therefore tested whether KC expression of UEX::HA was altered in flies with chronic alterations of cAMP metabolism, by introducing learning-relevant mutations in the *rutabaga*-encoded $Ca^{2+}$-stimulated adenylate cyclase, or the *dunce*-encoded cAMP-specific phosphodiesterase.

Anti-HA immunostaining of brains from $rut^{2080}$; *uex::HA* and $dnc^1$; *uex::HA* flies revealed a striking change in UEX localization (*Figure 7A and B* and *Videos 2–4*). Whereas UEX::HA is usually detected in the lobes of all KCs at a roughly equivalent level in wild-type flies, labeling was lower in the MB γ lobe and more pronounced in the $αβ_c$ KCs in $rut^{2080}$ and $dnc^1$ mutant backgrounds (*Figure 7C*), although the overall MB expression of UEX::HA is similar between wild-type and mutant flies (*Figure 7D*). In addition, western blot analyses of protein extracted from heads of these flies did not reveal a significant difference in overall UEX::HA expression levels (data not shown). These data are therefore consistent with cAMP regulating UEX function and perhaps its cellular localization in KCs.

## UEX is required to maintain a fluctuating $[Mg^{2+}]_i$ in αβ KCs

Although MagFRET can report $[Mg^{2+}]$ it does not respond quickly enough to record stimulus-evoked signals. We therefore constructed flies harboring UAS-transgenes for two newer genetically encoded $Mg^{2+}$ sensors, MagIC (non-FRET based; *Koldenkova et al., 2015*) and MARIO (FRET based; *Maeshima et al., 2018*). We were unable to detect UAS-MARIO expression in the fly brain and therefore could only use UAS-MagIC. MagIC was reported to respond most strongly to $Mg^{2+}$ but also to a lesser extent to $Ca^{2+}$ (*Koldenkova et al., 2015*). We therefore first verified the specificity of MagIC responses in a cell-permeabilized ex vivo fly brain preparation. Brains were removed from flies expressing UAS-MagIC in αβ KCs with c739-GAL4 (*Figure 8A*), incubated in a dish with saline (*Barnstedt et al., 2016*) and changes in fluorescence were monitored before and after bath application of chemicals. Whereas application of $MgCl_2$ evoked a dose-dependent increase in the MagIC response, chelation of $Mg^{2+}$ with EDTA produced a dose-dependent decrease (*Figure 8B* and *Videos 5 and 6*). In comparison, $CaCl_2$ only registered a slight increase at the highest concentrations whereas the more $Ca^{2+}$-selective chelator EGTA had little effect (*Figure 8B*). These results demonstrate that UAS-MagIC can monitor $[Mg^{2+}]_i$ in the αβ KCs in the fly brain.

Increasing intracellular cAMP has been shown to elicit $Mg^{2+}$ flux from mammalian cells (*Romani and Scarpa, 2000*; *Vormann and Günther, 1987*; *Jakob et al., 1989*; *Romani and Scarpa, 1990b*; *Romani and Scarpa, 1990a*; *Vormann and Günther, 1987*; *Günther et al., 1990*; *Howarth et al., 1994*). Since our experiments also indicated that cAMP might regulate UEX, we next tested whether stimulating cAMP synthesis with forskolin (FSK) might alter MagIC signals in αβ

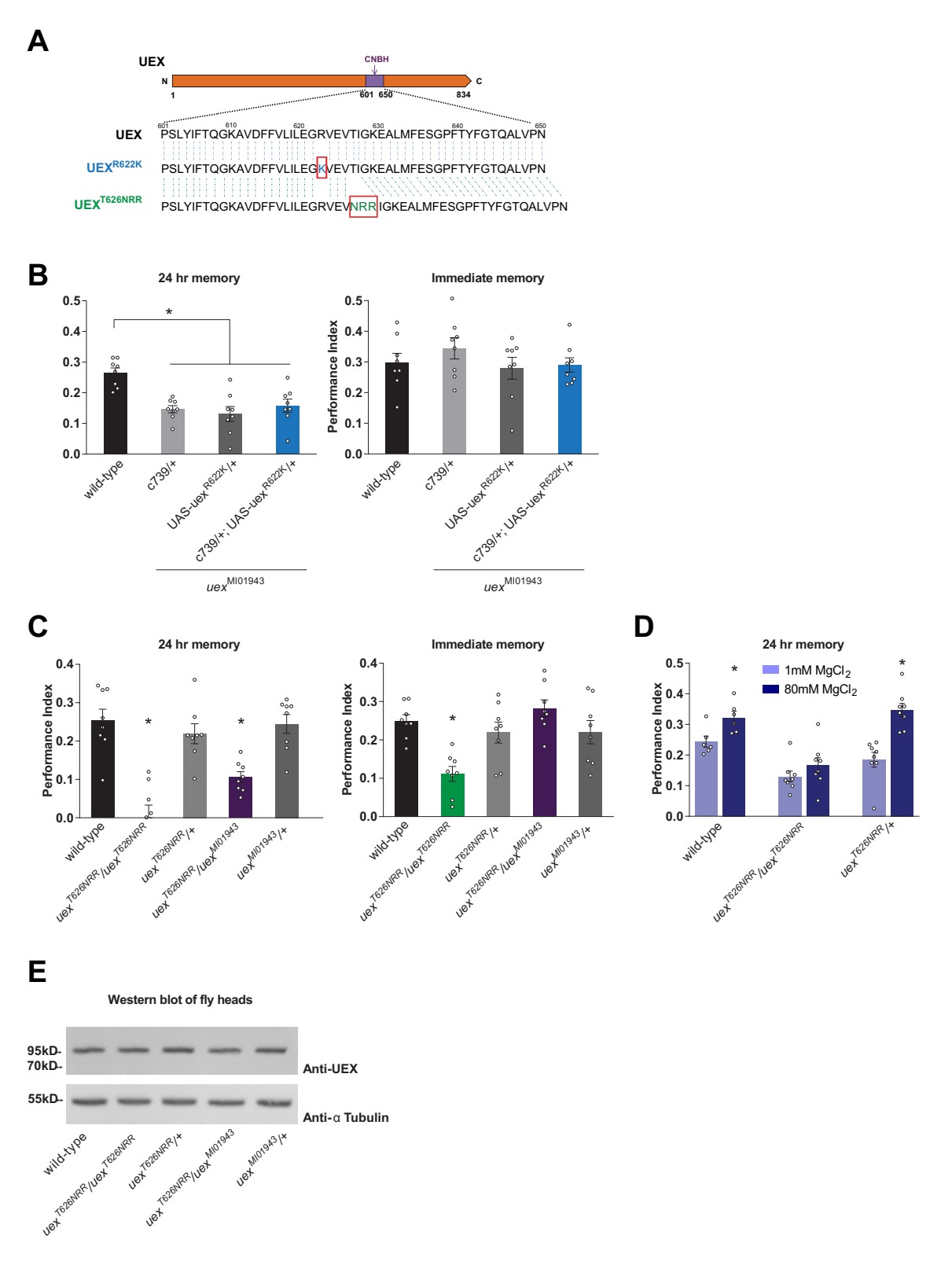

**Figure 6.** The cyclic nucleotide-binding homology (CNBH) domain of UEX is required for memory. (**A**) Schematic showing sequence detail of the CNBH domain in UEX, and the amino acid changes made in *uex*[R622K] and *uex*[T626NRR]. (**B**) Expressing a UAS-*uex*[R622K] transgene in αβ Kenyon cells did not rescue the LTM defect of *uex*[MI01943] mutant flies (p<0.05, ANOVA, n = 8). Immediate memory was also unaffected. (**C**) Flies homozygous for *uex*[T626NRR] have defective short- and long-term memory, while trans-heterozygous *uex*[T626NRR]/*uex*[MI01943] flies only exhibit impaired LTM (*p<0.05, ANOVA, n = 8). *Figure 6 continued on next page*

*Figure 6 continued*

(D) Dietary $Mg^{2+}$ did not enhance memory of homozygous $uex^{T626NRR}/uex^{T626NRR}$ flies (p<0.05, t-test, n = 8). (E) Western blot analysis of UEX protein expression in fly head extracts. Genotype from left to right: wild-type, $uex^{T626NRR}/uex^{T626NRR}$, $uex^{T626NRR}/+$, $uex^{T626NRR}/uex^{MI01943}$, $uex^{MI01943}/+$. The blot was first probed with anti-UEX antibody (upper panel), and then stripped and re-probed with anti-Tubulin antibody (lower panel) as a loading control.

KCs. For these experiments we again used an ex vivo brain preparation but this time the cells were not permeabilized. 30 μM FSK has been shown to evoke a peak increase in cAMP in KCs that approximates that observed following appetitive conditioning (*Louis et al., 2018*). Applying 30 μM FSK to *c739-GAL4; UAS-MagIC* brains evoked a consistent dynamic in MagIC fluorescence. After a sharp initial rise, responses slowly decayed back toward baseline before again rising slowly to a point at which the signal started to fluctuate. (*Figure 8C and D* and *Video 7*). The key signatures of this response were only recorded in the $Mg^{2+}$-sensitive Venus signal (*Figure 8D*). In contrast  mCherry fluorescence did not fluctuate but steadily decreased across the time course of the recording (likely a result of photo-bleaching), demonstrating that the fluctuation in the Venus signal is not a movement artifact (*Figure 8E*). Importantly, FSK induced MagIC responses were greater than those following application of saline (*Figure 8—figure supplement 1A*). However, a fluctuating response also developed after saline applications (*Figure 8—figure supplement 1B*) suggesting that the rhythmic MagIC signal may be a general response to an increase in $[Mg^{2+}]_i$ that follows cellular perturbation.

The *Drosophila* MB has previously been reported to exhibit a slow (0.004 Hz) $Ca^{2+}$ oscillation in ex vivo brains whereas a much faster 20 Hz oscillation is evoked by odors in the locust MB (*Laurent and Naraghi, 1994*; *Rosay et al., 2001*). Although our initial characterization of MagIC in the fly brain indicated a preferential response to $Mg^{2+}$ (*Figure 8B*), we nevertheless explicitly tested whether FSK  induced fluctuation of the $[Ca^{2+}]_i$ of αβ KCs, using expression of UAS-GCaMP6f (*Chen et al., 2013*). FSK induced a delayed increase in the GCaMP response but no clear oscillatory activity was observed (*Figure 8—figure supplement 1C–E*).

Lastly, we tested whether the observed MagIC responses were sensitive to the status of the *uex* gene. We generated $uex^{MI01943}$ flies that also harbored c379-GAL4 and UAS-MagIC and compared their FSK- and saline-induced MagIC responses to those of flies with a wild-type *uex* locus. The $uex^{MI01943}$ mutant flies showed an increased FSK response to that of wild-type flies, whereas saline-evoked responses were indistinguishable (*Figure 8F and G*). Responses evoked by the inactive FSK analogue, ddFSK, were also insensitive to the status of *uex* (*Figure 8—figure supplement 1F*). Mutation of *uex* therefore selectively increases mean FSK-evoked MagIC responses.

We also noticed that MagIC traces from *uex* mutant flies did not exhibit a fluctuating signal (*Figure 8H* and *Figure 8—figure supplement 1G*). To quantify this difference we calculated the mean power spectral density (PSD) of traces from $uex^{MI01943}$ and wild-type flies treated with FSK or saline. In both conditions the mean PSD was significantly left-shifted toward lower frequencies in the $uex^{MI01943}$ mutants compared to the wild-type controls (*Figure 8I*). Wild-type fly brains had significantly more oscillatory activity centered around 0.015 Hz than those from $uex^{MI01943}$ mutants. These data therefore suggest that UEX is required for slow rhythmic maintenance of KC $[Mg^{2+}]_i$. Importantly, finding that MagIC signals are elevated and altered in *uex* mutants confirms that the observed MagIC responses are $Mg^{2+}$-dependent. Moreover, they suggest that the KC expressed UEX limits $Mg^{2+}$ accumulation, consistent with a role in extrusion.

## Discussion

We observed an enhancement of olfactory LTM performance when flies were fed for 4 days before training with food supplemented with 80 mM $[Mg^{2+}]$. This result resembles that reported in rats, although longer periods of feeding were required to raise brain $[Mg^{2+}]$ to memory-enhancing levels (*Slutsky et al., 2010*). A difference in optimal feeding time may reflect the size of the animal and perhaps the greater bioavailability of dietary $Mg^{2+}$ in *Drosophila*. Whereas $Mg^{2+}$-L-threonate (MgT) was a more effective means of delivering $Mg^{2+}$ than magnesium chloride in rats (*Slutsky et al., 2010*), we observed a similar enhancement of memory performance when flies were fed with magnesium chloride, magnesium sulfate, or MgT (data not shown).

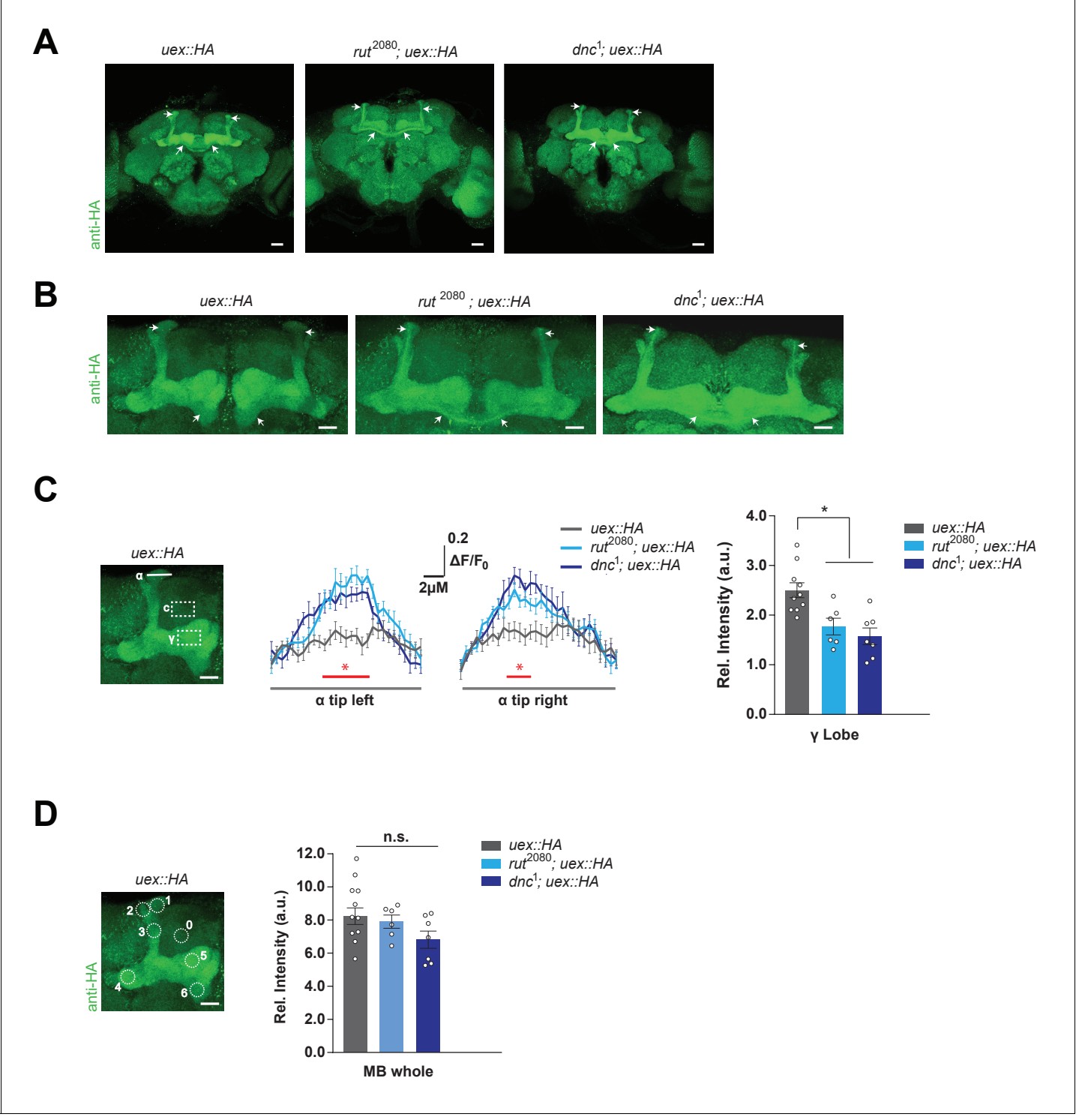

**Figure 7.** Kenyon cell (KC) *uex* expression is altered in *rutabaga* and *dunce* mutant flies. (**A**) Anti-HA stained brains reveal UEX::HA protein localization is altered in *rut*[2080]; *uex::HA* and *dnc*[1]; *uex::HA* flies, becoming more prominent in αβ$_c$ KCs (arrows). Scale bars 20 µm. (**B**) Enlarged images of the mushroom bodies (MBs) highlighting αβ$_c$ KC expression in *rut*[2080] and *dnc*[1] mutant flies, as compared with wild-type *uex::HA* flies. Scale bars 20 µm. (**C**) Quantification of fluorescence intensity. Left, micrograph with a measurement line through the α lobe tip and rectangular ROIs for the γ lobe and a control area. Middle, relative fluorescence intensity profiles across the α lobe tip show significantly higher signal in *rut*[2080] and *dnc*[1] mutant flies in the center region occupied by the αβ core KCs (*p<0.05, ANOVA, n = 6–10). Right, the relative intensity in the γ lobe was significantly lower in *rut*[2080] and *dnc*[1] mutant flies, as compared to wild-type controls (*p<0.05, ANOVA, n = 6–10). Scale bars 10 µm. (**D**) Left, micrograph showing circular ROIs. Right,

*Figure 7 continued on next page*

*Figure 7 continued*

quantification. Total intensity over all six ROIs on the MBs was not significantly different between the $rut^{2080}$, $dnc^1$ and wild-type brains (p>0.13; ANOVA, n = 6–10).

Elevating $[Mg^{2+}]_e$ in the rat brain leads to a compensatory upregulation of expression of the NR2B subunit of the NMDAR and therefore an increase in the proportion of postsynaptic NR2B-containing NMDARs. This class of NMDARs have a longer opening time (*Chen et al., 1999*; *Erreger et al., 2005*) suggesting that this switch in subunit composition represents a homeostatic plasticity mechanism (*Turrigiano, 2008*) to accommodate for the increased NMDAR block imposed by increasing $[Mg^{2+}]_e$. Moreover, overexpression of NR2B in the mouse forebrain can enhance synaptic facilitation and learning and memory performance (*Tang et al., 1999*), supporting an increase in NR2B being an important factor in $Mg^{2+}$-enhanced memory. However, even in the original in vitro study of $Mg^{2+}$-enhanced synaptic plasticity (*Slutsky et al., 2004*), it was noted that NMDAR currents were insufficient to fully explain the observed changes.

Our NMDAR subunit loss-of-function studies in the *Drosophila* KCs did not impair regular or $Mg^{2+}$-enhanced memory. Furthermore, we did not detect an obvious change in the levels of brain-wide expression of glutamate receptor subunits in $Mg^{2+}$-fed flies. Although NMDAR activity has previously been implicated in *Drosophila* olfactory memory, the effects were mostly ascribed to function outside the MB (*Xia et al., 2005*; *Wu et al., 2007*). In addition, overexpressing *Nmdar1* in all neurons, or specifically in all KCs, did not alter STM or LTM. Ectopic overexpression in the MB of an NMDAR$^{N631Q}$ version, which cannot be blocked by $Mg^{2+}$, impaired LTM (*Miyashita et al., 2012*). However, this mutation permits ligand-gated $Ca^{2+}$ entry, without the need for correlated neuronal depolarization, which may perturb KC function in unexpected ways. It is perhaps most noteworthy that learning-relevant synaptic depression in the MB can be driven by dopaminergic teaching signals delivered to cholinergic output synapses from odor-responsive KCs to specific MBONs (*Claridge-Chang et al., 2009*; *Aso et al., 2012*; *Burke et al., 2012*; *Liu et al., 2012*; *Owald et al., 2015*; *Hige et al., 2015*; *Barnstedt et al., 2016*; *Perisse et al., 2016*; *Aso et al., 2014*; *Owald and Waddell, 2015*; *Handler et al., 2019*). It is conceivable that KCs receive glutamate, from a source yet to be identified, but there is currently no obvious place in the MB network for NMDAR-dependent plasticity. Evidence therefore suggests that normal and $Mg^{2+}$-enhanced *Drosophila* LTM is independent of NMDAR signaling in KCs. In addition, our MagFRET measurements indicate that $Mg^{2+}$ feeding also increases the $[Mg^{2+}]_i$ of αβ KCs by approximately 50 μM.

We identified a role for *uex*, the single fly ortholog of the evolutionarily conserved family of CNNM-type $Mg^{2+}$ efflux transporters (*Ishii et al., 2016*). There are four distinct *CNNM* genes in mice and humans, five in *C. elegans*, and two in zebrafish (*Ishii et al., 2016*; *Arjona et al., 2013*). The *uex* locus produces four alternatively spliced mRNA transcripts, but all encode the same 834 aa protein. The precise role of CNNM proteins in $Mg^{2+}$ transport is somewhat contentious (*Funato et al., 2018a*; *Arjona and de Baaij, 2018*; *Funato et al., 2018b*; *Giménez-Mascarell et al., 2019*). Some propose that CNNM proteins are direct $Mg^{2+}$ transporters, whereas others favor that they function as sensors of intracellular $Mg^{2+}$ concentration $[Mg^{2+}]_i$ and/or regulators of other $Mg^{2+}$ transporters. We found that ectopic expression of *Drosophila* UEX enhances $Mg^{2+}$ efflux in HEK293 cells and that endogenous UEX limits $[Mg^{2+}]_i$ in αβ KCs in the fly brain. Therefore, if UEX is not itself a $Mg^{2+}$ transporter, it must be

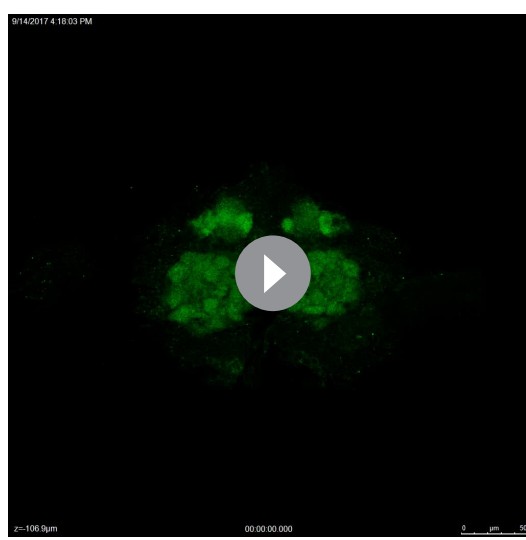

**Video 2.** Expression of UEX in a wild-type *Drosophila* brain. Confocal Z-stack of a *uex::HA* fly brain stained with anti-HA antibody.
https://elifesciences.org/articles/61339#video2

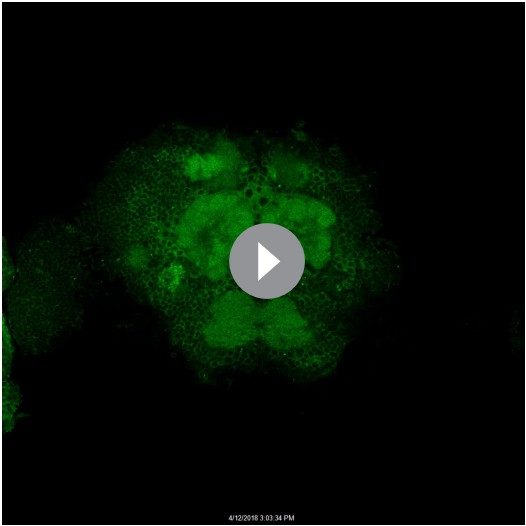
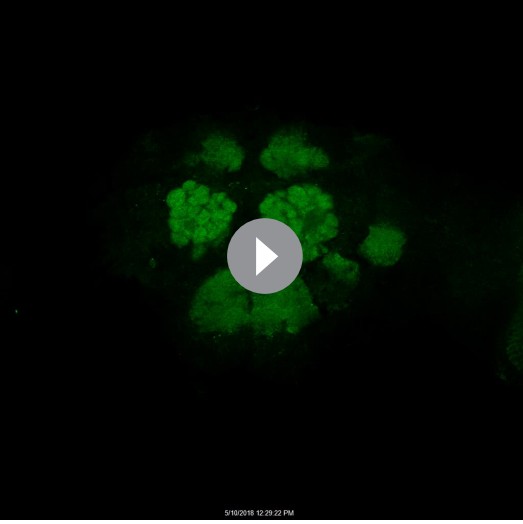

**Video 3.** Expression of UEX in a *rut²⁰⁸⁰Drosophila* brain. Confocal Z-stack of a brain from a *rut²⁰⁸⁰; uex:: HA* fly stained with anti-HA antibody. The αβ_c Kenyon cells label more prominently than in the wild-type *uex:: HA* brain in **Video 2**.
https://elifesciences.org/articles/61339#video3

**Video 4.** Expression of UEX in a *dnc¹Drosophila* brain. Confocal Z-stack of a brain from a *dnc¹; uex::HA* fly stained with anti-HA antibody. The αβ_c Kenyon cells label more prominently than in the wild-type *uex::HA* brain in **Video 2**.
https://elifesciences.org/articles/61339#video4

able to interact effectively with human $Mg^{2+}$ efflux transporters and to influence $Mg^{2+}$ extrusion in *Drosophila*. Since UEX is the only CNNM protein in the fly, it may serve all the roles of the four individual mammalian CNNMs. However, the ability of mouse CNNM2 to restore memory capacity to *uex* mutant flies suggests that the memory-relevant UEX function can be substituted by that of CNNM2.

Interestingly, none of the disease-relevant variants of CNNM2 were able to complement the memory defect of *uex* mutant flies. The CNNM2 T568I variant substitutes a single amino acid in the second CBS domain (*Arjona et al., 2014*). The oncogenic protein tyrosine phosphatases of the PRL (phosphatase of regenerating liver) family bind to the CBS domains of CNNM2 and CNNM3 and can inhibit their $Mg^{2+}$ transport function (*Hardy et al., 2015*; *Giménez-Mascarell et al., 2017*; *Zhang et al., 2017*). It will therefore be of interest to test the role of the UEX CBS domains and whether fly PRL-1 regulates UEX activity.

RNA-seq analysis reveals that *uex* is strongly expressed in the larval and adult fly digestive tract and nervous systems, as well as the ovaries (*Gelbart and Emmert, 2010*; *Croset et al., 2018*; *Davie et al., 2018*) suggesting that many *uex* mutations will be pleiotropic. Our *uexΔ* allele, which deletes 272 amino acids (including part of the second CBS and the entire CNBH domain) from the UEX C-terminus, results in developmental lethality when homozygous, demonstrating that *uex* is an essential gene. Mammalian CNNM4 is localized to the basolateral membrane of intestinal epithelial cells (*Yamazaki et al., 2013*). There it is believed to function in transcellular $Mg^{2+}$ transport by exchanging intracellular $Mg^{2+}$ for extracellular $Na^+$ following apical entry through TRPM7 channels. Lethality in *Drosophila* could therefore arise from an inability to absorb sufficient $Mg^{2+}$ through the larval gut. However, neuronally restricted expression of *uex*^RNAi with *elav*-GAL4 also results in larval lethality (data not shown), suggesting UEX has an additional role in early development of the nervous system, like CNNM2 in humans and zebrafish (*Arjona et al., 2014*; *Accogli et al., 2019*). Perhaps surprisingly, flies carrying homozygous or *trans*-heterozygous combinations of several hypomorphic *uex* alleles have defective appetitive and aversive memory performance, yet they seem otherwise unaffected.

Genetically engineering the *uex* locus to add a C-terminal HA tag to the UEX protein allowed us to localize its expression in the brain. Labeling is particularly prominent in all major classes of KCs. Restricting knockdown of *uex* expression to all αβ KCs of adult flies, or even just the αβ_c subset reproduced the LTM defect. The LTM impairment was evident if *uex*^RNAi expression in αβ neurons

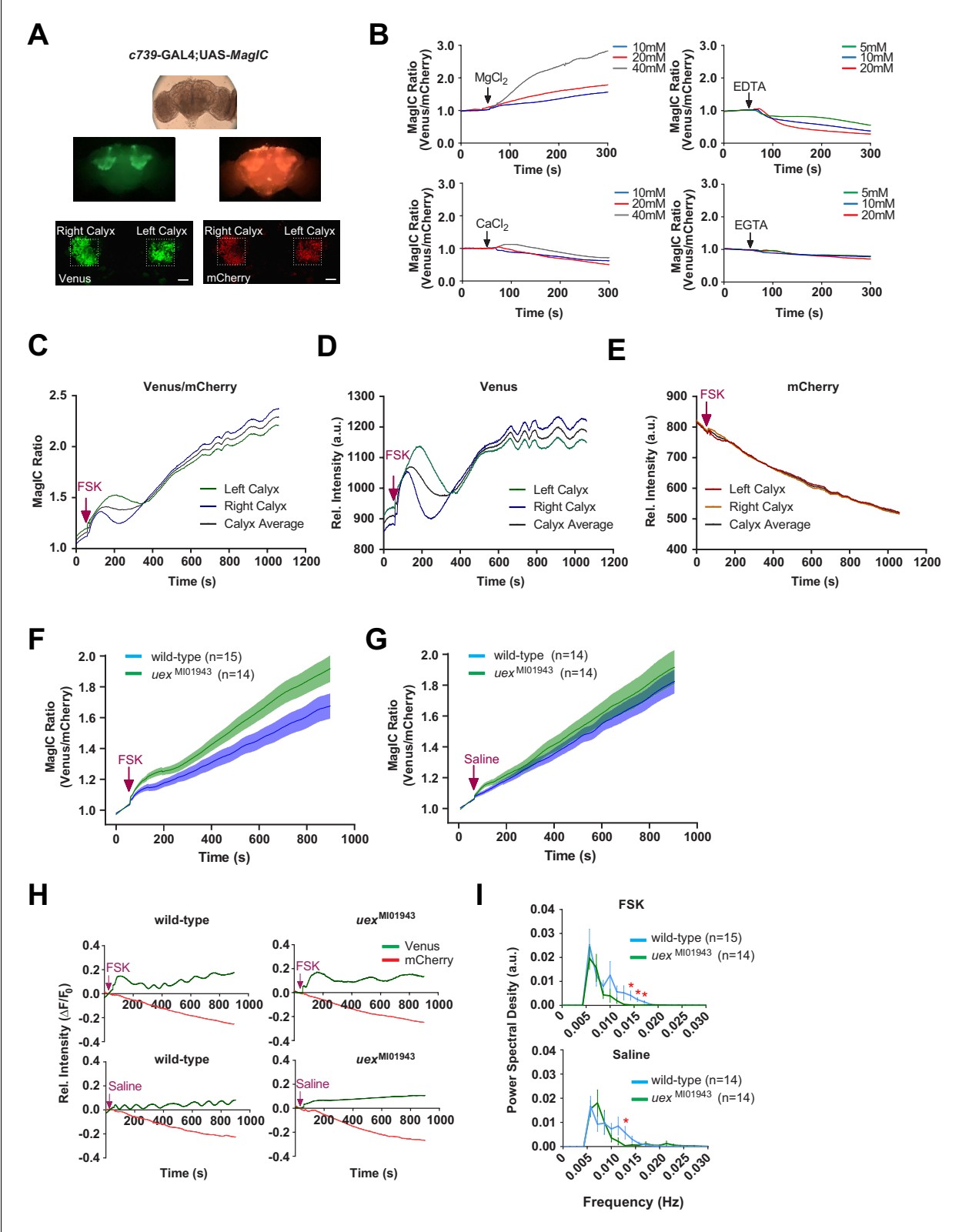

**Figure 8.** UEX limits a rise in [Mg$^{2+}$]$_i$ and supports a slow oscillation in αβ Kenyon cells (KCs). (**A**) Explant fly brain expressing UAS-MagIC driven by *c739*-GAL4. Upper panel, wide-field phase contrast view; middle panels, fluorescence views of Venus and mCherry channels; lower panel, confocal section at the level of the KC somata showing Venus and mCherry channels. Scale bars 20 μm. (**B**) MagIC selectively responds to changes in [Mg$^{2+}$]$_i$ in KCs. Traces of MagIC ratio following bath application of 10, 20, or 40 mM MgCl$_2$ or CaCl$_2$; 5, 10, or 20 mM EDTA or EGTA. (**C**) Representative trace of
*Figure 8 continued on next page*

*Figure 8 continued*

MagIC ratio following application of FSK shows an initial wave followed by a gradual rise and the development of a slow oscillation. (D) The primary responses result from changes in the $Mg^{2+}$-sensitive Venus signal. (E) The mCherry signal exhibits a steady decay. (F) FSK-evoked MagIC responses are greater in *uex* mutant flies. Averaged MagIC responses show that FSK induced a significantly greater increase in $uex^{MI01943}$ mutant than in wild-type flies. (G). Averaged saline-evoked MagIC responses were not significantly altered in *uex* mutant flies. (H) Individual Venus (green) and mCherry (red) channel traces showing that the slow oscillation is only evident in the Venus channel of wild-type, but not *uex* mutant, flies. (I) Power spectral density (PSD) analysis of the time series from 200 to 900 s of all data shows that traces from wild-type flies have significantly more oscillatory activity, centered around 0.015 Hz, than those from *uex* mutant flies.

The online version of this article includes the following figure supplement(s) for figure 8:

**Figure supplement 1.** Individual traces for MagIC and GCaMP imaging.

was restricted to adult flies, suggesting UEX has a more sustained role in neuronal physiology. In contrast, knocking down *uex* expression in either the $\alpha\beta_s$ or $\alpha'\beta'$ neurons did not impair LTM. Activity of $\alpha'\beta'$ neurons is required after training to consolidate appetitive LTM (**Krashes and Waddell, 2008**), whereas $\alpha\beta_c$ and $\alpha\beta_s$ KC output, together and separately, is required for its expression (**Krashes and Waddell, 2008**; **Perisse et al., 2013**). Therefore, observing normal LTM performance in flies with *uex* loss-of-function in $\alpha\beta_s$ and $\alpha'\beta'$ neurons argues against a general deficiency of $\alpha\beta$ neuronal function when manipulating *uex*.

Dietary $Mg^{2+}$ could not enhance the defective LTM performance of flies that were constitutively *uex* mutant, or harbored $\alpha\beta$ KC-restricted *uex* loss-of-function. However, expressing *uex* in the $\alpha\beta$ KCs of *uex* mutant flies restored the ability of $Mg^{2+}$ to enhance performance. Therefore, the $\alpha\beta$ KCs are the cellular locus for $Mg^{2+}$-enhanced memory in the fly.

It perhaps seems counterintuitive that UEX-directed magnesium efflux is required in KCs to support the memory-enhancing effects of $Mg^{2+}$ feeding, when dietary $Mg^{2+}$ elevates KC $[Mg^{2+}]_i$. At this stage, we can only speculate as to why this is the case. We assume that the brain and $\alpha\beta$ KCs, in particular, have to adapt in a balanced way to the higher levels of intracellular and extracellular $Mg^{2+}$ that result from dietary supplementation. Our live-imaging of KC $[Mg^{2+}]_i$ in wild-type and *uex* mutant brains suggests that UEX-directed efflux is likely to be an essential factor in the active, and perhaps stimulus-evoked, homeostatic maintenance of these elevated levels.

A number of mammalian cell-types extrude $Mg^{2+}$ in a cAMP-dependent manner, a few minutes after being exposed to β-adrenergic stimulation (**Romani and Scarpa, 2000**; **Vormann and Günther, 1987**; **Jakob et al., 1989**; **Romani and Scarpa, 1990b**; **Romani and Scarpa, 1990a**; **Vormann and Günther, 1987**; **Günther et al., 1990**; **Howarth et al., 1994**). The presence of a CNBH domain suggests that UEX and CNNMs could be directly regulated by cAMP. We tested the importance of the CNBH by introducing an R622K amino acid substitution that should block cAMP binding in the UEX CNBH. This subtle mutation abolished the ability of the $uex^{R622K}$ transgene to restore LTM performance to *uex* mutant flies. We also used CRISPR to mutate the CNBH in the native *uex* locus. Although deleting the CNBH from CNNM4 abolished $Mg^{2+}$ efflux activity (**Chen et al., 2018**), flies homozygous for the $uex^{T626NRR}$ lesion were viable, demonstrating that they retain a sufficient level of UEX function. However, these flies exhibited impaired immediate and long-term memory. In addition, the performance of $uex^{T626NRR}$ flies could not be enhanced by $Mg^{2+}$ feeding. These data demonstrate that an intact CNBH is a critical element of memory-relevant UEX function. Binding of clathrin adaptor proteins to the CNNM4 CNBH has been implicated in basolateral targeting (**Hirata et al., 2014**), suggesting that $UEX^{T626NRR}$ might be inappropriately localized in KCs. Furthermore, KC expression of the CNNM2 E122K mutant variant, which retains residual function

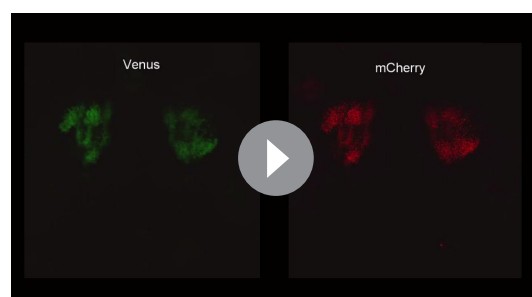

**Video 5.** KC-expressed MagIC responds to $Mg^{2+}$ application. Confocal time-series recording from a c739/+; UAS-MagIC/+ fly brain shows an increase in Venus, but not mCherry, fluorescence signal in response to 20 mM $MgCl_2$ application.
https://elifesciences.org/articles/61339#video5

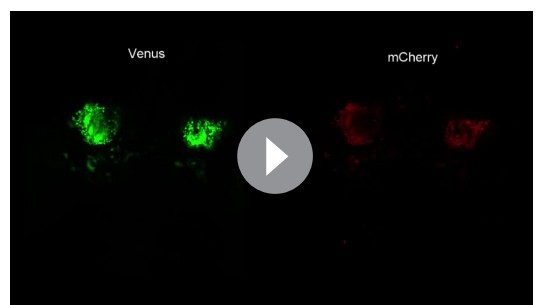

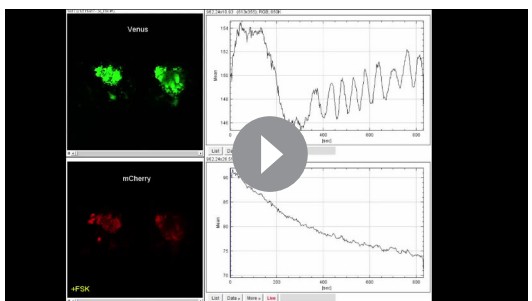

**Video 6.** KC-expressed MagIC responds to $Mg^{2+}$ chelation. Confocal time-series recording from a c739/+; UAS-MagIC/+ fly brain shows a strong decrease in Venus and a weak decrease in mCherry fluorescence signal in response to 10 mM EDTA application. https://elifesciences.org/articles/61339#video6

**Video 7.** KC-expressed MagIC reveals slow oscillation of intracellular $Mg^{2+}$. Confocal time-series recording from a c739/+; UAS-MagIC/+ fly brain shows a slow oscillation in Venus, but not mCherry, fluorescence signal in response to 30 μM forskolin. https://elifesciences.org/articles/61339#video7

but has a trafficking defect (*Arjona et al., 2014*), did not restore the *uex* LTM defect.

Although it has been questioned whether the CNNM2/3 CNBH domains bind cyclic nucleotides (*Chen et al., 2018*), we found that FSK evoked an increase in αβ KC $[Mg^{2+}]_i$ that was sensitive to *uex* mutation, and that UEX::HA was mislocalized in $rut^{2080}$ adenylate cyclase (*Han et al., 1992*) and $dnc^1$ phosphodiesterase (*Dudai et al., 1976*) learning defective mutant flies. Whereas UEX::HA label was evenly distributed in γ, $αβ_c$, and $αβ_s$ KCs in wild-type flies, UEX::HA label was diminished in the γ and $αβ_s$ KCs and was stronger in $αβ_c$ neurons in $rut^{2080}$ and $dnc^1$ mutants. The chronic manipulations of cAMP in the mutants are therefore consistent with cAMP impacting UEX localization, perhaps by interacting with the CNBH. In addition, altered UEX localization may contribute to the memory defects of $rut^{2080}$ and $dnc^1$ flies.

Our physiological data using Magnesium Green in mammalian cell culture and the genetically encoded MagIC reporter in αβ KCs demonstrate that fly UEX facilitates $Mg^{2+}$ efflux. Stimulating the fly brain with FSK evoked a greater increase in αβ KC $[Mg^{2+}]_i$ in *uex* mutant brains than in wild-type controls which provides the first evidence that UEX limits a rise in $[Mg^{2+}]_i$ in *Drosophila* KCs. Our MagIC recordings also revealed a slow oscillation (centered around 0.015 Hz, approximately once a minute) of αβ KC $[Mg^{2+}]_i$ that was dependent on UEX. We do not yet understand the physiological function of this $[Mg^{2+}]_i$ fluctuation although it likely reflects a homeostatic systems-level property of the cells. Biochemical oscillatory activity plays a crucial role in many aspects of cellular physiology (*Novák and Tyson, 2008*). Most notably, circadian timed fluctuation of $[Mg^{2+}]_i$ links dynamic cellular energy metabolism to clock-controlled translation through the $Mg^{2+}$ sensitive mTOR (mechanistic target of rapamycin) pathway (*Feeney et al., 2016*). It is therefore possible that slow $Mg^{2+}$ oscillations could unite roles for cAMP, UEX, energy flux (*Plaçais et al., 2017*), and mTOR-dependent translation underlying LTM-relevant synaptic plasticity (*Casadio et al., 1999*; *Huber et al., 2000*; *Beaumont et al., 2001*; *Hou and Klann, 2004*; *Hoeffer et al., 2008*).

# Materials and methods

## Key resources table

| Reagent type (species) or resource | Designation | Source or reference | Identifiers | Additional information |
|---|---|---|---|---|
| Genetic reagent (*Drosophila melanogaster*) | Canton-S | Originally from W.G.Quinn lab | Canton-S | Waddell Lab stock |
| Genetic reagent (*D. melanogaster*) | UAS-*EGFP* | Bloomington *Drosophila* Stock Center | RRID:BDSC_5431 | |

*Continued on next page*

*Continued*

| Reagent type (species) or resource | Designation | Source or reference | Identifiers | Additional information |
|---|---|---|---|---|
| Genetic reagent (*D. melanogaster*) | c739-GAL4 | *McGuire et al., 2001* | c739-GAL4 | Lab stock |
| Genetic reagent (*D. melanogaster*) | c305a-GAL4 | *Krashes et al., 2007* | c305a-GAL4 | Lab stock |
| Genetic reagent (*D. melanogaster*) | NP7175-GAL4 | *Tanaka et al., 2004* | NP7175-GAL4 | Lab stock |
| Genetic reagent (*D. melanogaster*) | 0770-GAL4 | *Gohl et al., 2011* | 0770-GAL4 | Lab stock |
| Genetic reagent (*D. melanogaster*) | MB247-GAL4 | *Zars et al., 2000* | MB247-GAL4 | Lab stock |
| Genetic reagent (*D. melanogaster*) | *nSyb*-GAL4 | Bloomington *Drosophila* Stock Center | RRID:BDSC_51635 | Gift from J. Simpson |
| Genetic reagent (*D. melanogaster*) | *elav*-GAL4 | Bloomington *Drosophila* Stock Center | RRID:BDSC_8765 | |
| Genetic reagent (*D. melanogaster*) | *tub*PGAL80$^{ts}$ | *McGuire et al., 2003* | tubP-GAL80$^{ts}$ | Lab stock |
| Genetic reagent (*D. melanogaster*) | UAS-*Nmdar1*$^{RNAi}$ | Bloomington *Drosophila* Stock Center | RRID:BDSC_25941 | |
| Genetic reagent (*D. melanogaster*) | *nos*-Cas9.P | Bloomington *Drosophila* Stock Center | RRID:BDSC_54591 | |
| Genetic reagent (*D. melanogaster*) | *nos*-Cas9(X) | Fly Stocks of National Institute of Genetics | CAS0002 | |
| Genetic reagent (*D. melanogaster*) | *lig4* KO *vasa*-Cas9 | *Zimmer et al., 2016* | *lig4* KO vasa-Cas9 | Gift from C. Zimmer |
| Genetic reagent (*D. melanogaster*) | PhsILMiT | Bloomington *Drosophila* Stock Center | RRID:BDSC_24613 | |
| Genetic reagent (*D. melanogaster*) | *rut*$^{2080}$ | *Han et al., 1992* | *rut*$^{2080}$ | Lab stock |
| Genetic reagent (*D. melanogaster*) | *dnc*$^{1}$ | *Dudai et al., 1976* | *dnc*$^{1}$ | Lab stock |
| Genetic reagent (*D. melanogaster*) | *uex*$^{MI01943}$ | Bloomington *Drosophila* Stock Center | RRID:BDSC_32805 | |
| Genetic reagent (*D. melanogaster*) | *uex*$^{NC1}$ | Bloomington *Drosophila* Stock Center | RRID:BDSC_7176 | |
| Genetic reagent (*D. melanogaster*) | UAS-*uex*$^{RNAi}$ | Bloomington *Drosophila* Stock Center *Perkins et al., 2015* | RRID:BDSC_36116 | |

*Continued on next page*

*Continued*

| Reagent type (species) or resource | Designation | Source or reference | Identifiers | Additional information |
|---|---|---|---|---|
| Genetic reagent (*D. melanogaster*) | *uex*-GAL4 | Vienna *Drosophila* Resource Center | VT23256 | |
| Genetic reagent (*D. melanogaster*) | UAS-GCaMP6f | Bloomington *Drosophila* Stock Center | RRID:BDSC_42747 | |
| Genetic reagent (*D. melanogaster*) | *uex*$^{MI01943.ex1}$ | This study | *uex*$^{MI01943.ex1}$ | See Methods and *Figure 2—figure supplement 2A and B* |
| Genetic reagent (*D. melanogaster*) | *uex*$^{MI01943.ex2}$ | This study | *uex*$^{MI01943.ex2}$ | See Methods and *Figure 2—figure supplement 2A and B* |
| Genetic reagent (*D. melanogaster*) | *uex*Δ | This study | *uex*Δ | See Methods and *Figure 2—figure supplement 2C* |
| Genetic reagent (*D. melanogaster*) | *uex::HA* | This study | *uex::HA* | See Methods and *Figure 3—figure supplement 1* |
| Genetic reagent (*D. melanogaster*) | *uex*$^{T626NRR}$ | This study | *uex*$^{T626NRR}$ | See Methods and *Figure 6A* |
| Genetic reagent (*D. melanogaster*) | UAS-*uex* | This study | UAS-*uex* | See Methods |
| Genetic reagent (*D. melanogaster*) | UAS-*uex*$^{R622K}$ | This study | UAS-*uex*$^{R622K}$ | See Methods and *Figure 6A* |
| Genetic reagent (*D. melanogaster*) | UAS-*CNNM2*$^{WT}$ | This study | UAS-*CNNM2*$^{WT}$ | See Methods and *Figure 5A* |
| Genetic reagent (*D. melanogaster*) | UAS-*CNNM2*$^{E357K}$ | This study | UAS-*CNNM2*$^{E357K}$ | See Methods and *Figure 5A* |
| Genetic reagent (*D. melanogaster*) | UAS-*CNNM2*$^{T568I}$ | This study | UAS-*CNNM2*$^{T568I}$ | See Methods and *Figure 5A* |
| Genetic reagent (*D. melanogaster*) | UAS-*CNNM2*$^{S269W}$ | This study | UAS-*CNNM2*$^{S269W}$ | See Methods and *Figure 5A* |
| Genetic reagent (*D. melanogaster*) | UAS-*CNNM2*$^{E122K}$ | This study | UAS-*CNNM2*$^{E122K}$ | See Methods and *Figure 5A* |
| Genetic reagent (*D. melanogaster*) | UAS-*MagFRET-1* | This study | UAS-*MagFRET-1* | See Methods |
| Genetic reagent (*D. melanogaster*) | UAS-*MARIO* | This study | UAS-*MARIO* | See Methods |
| Genetic reagent (*D. melanogaster*) | UAS-*MagIC* | This study | UAS-*MagIC* | See Methods |
| Antibody | Anti-GFP (Rabbit polyclonal) | Invitrogen | Cat# A-11122, RRID:AB_221569 | IF (1:250) |
| Antibody | Anti-HA (Rabbit monoclonal) | New England Biolabs | Cat# 3724T | IF (1:250) |
| Antibody | Anti-FLAG (Rabbit polyclonal) | Sigma-Aldrich | Cat# F-7425, RRID:AB_439687 | IF (1:250) |
| Antibody | Anti-UEX (Rabbit polyclonal) | Eurogentec | Cat# ZGB-15047 | WB (1:2000) |
| Antibody | Anti-Tubulin (Mouse monoclonal) | Sigma-Aldrich | Cat# T-6199, RRID:AB_477583 | WB (1:2000) |
| Antibody | Anti-rabbit IgG (Alexa 488 goat polyclonal) | Invitrogen | Cat# A-11034, RRID:AB_2576217 | IF (1:250) |

*Continued on next page*

*Continued*

| Reagent type (species) or resource | Designation | Source or reference | Identifiers | Additional information |
|---|---|---|---|---|
| Antibody | Anti-rabbit IgG (HRP-conjugated goat polyclonal) | Thermo Fisher | Cat# 32260, RRID:AB_1965959 | WB (1:5000) |
| Antibody | Anti-mouse IgG (HRP-conjugated goat polyclonal) | Thermo Fisher | Cat# 32230, RRID:AB_1965958 | WB (1:5000) |
| Recombinant DNA reagent | pUAST-*uex* (plasmid) | This study | | pUAST vector containing *uex* cDNA |
| Recombinant DNA reagent | pUAST- *uex*$^{R622K}$ (plasmid) | This study | | pUAST vector containing *uex*$^{R622K}$ cDNA |
| Recombinant DNA reagent | pUAST- *CNNM2*$^{WT}$ (plasmid) | This study | | pUAST vector containing mouse *CNNM2*$^{WT}$ cDNA |
| Recombinant DNA reagent | pUAST- *CNNM2*$^{E122K}$ (plasmid) | This study | | pUAST vector containing mouse *CNNM2*$^{ET22K}$ cDNA |
| Recombinant DNA reagent | pUAST- *CNNM2*$^{E357K}$ (plasmid) | This study | | pUAST vector containing mouse *CNNM2*$^{E357K}$ cDNA |
| Recombinant DNA reagent | pUAST- *CNNM2*$^{S269W}$ (plasmid) | This study | | pUAST vector containing mouse *CNNM2*$^{S269W}$ cDNA |
| Recombinant DNA reagent | pUAST- *CNNM2*$^{T568I}$ (plasmid) | This study | | pUAST vector containing mouse *CNNM2*$^{T568I}$ cDNA |
| Recombinant DNA reagent | pJFRC-MUH-MagFRET-1 (plasmid) | This paper | | pJFRC-MUH vector containing MagFRET-1 CDS |
| Recombinant DNA reagent | pTW-MARIO (plasmid) | This paper | | pTW vector containing MARIO CDS |
| Recombinant DNA reagent | pTW-MagIC (plasmid) | This paper | | pTW vector containing MagIC CDS |
| Recombinant DNA reagent | pCFD3-dU6:3gRNA vector | Addgene | RRID:Addgene_49410 | |
| Recombinant DNA reagent | pCMVMagFRET-1 | Addgene | RRID:Addgene_50742 | |
| Recombinant DNA reagent | pScarlessHD-2xHA-DsRed | Addgene | 80822 | Gift to Addgene from Kate O'Connor-Giles |
| Recombinant DNA reagent | gRNA constructs for *uexΔ* | GenetiVision | Y17.C253.Q002 | Generated by GenetiVision for this study |
| Recombinant DNA reagent | Donor construct for *uexΔ* | GenetiVision | Y17.C253.Q002 | Generated by GenetiVision for this study |
| Recombinant DNA reagent | gRNA construct for *uex::HA* | WellGenetics | WG-16107 gRNA | Generated by WellGenetics for this study |
| Recombinant DNA reagent | Donor construct for *uex::HA* | WellGenetics | PWG1521 pUC57-Kan-16107 donor | Generated by WellGenetics for this study |
| Recombinant DNA reagent | gRNA construct for *uex*$^{T626NRR}$ | This study | | |

*Continued on next page*

*Continued*

| Reagent type (species) or resource | Designation | Source or reference | Identifiers | Additional information |
|---|---|---|---|---|
| Sequence-based reagent | Gipc1_F | This study | PCR primers | GGGAAAGGAC AAAAGGAACCC |
| Sequence-based reagent | *uex* CDS, Forward | This study | PCR primers | ATCGCCGCGGAT GAACACATATTT CATATCATTTATTAC |
| Sequence-based reagent | *uex* CDS, Reverse | This study | PCR primers | ATCGCTCGAGTTA GGGCTTACTTT GCTTGCTC |
| Sequence-based reagent | *uex*$^{R622K}$, fragment 1, Forward | This study | PCR primers | ATGAACACATATTT CATATCATTTAT TACAATAATTA |
| Sequence-based reagent | *uex*$^{R622K}$, fragment 1, Reverse | This study | PCR primers | GTGACTTCTACT TTACCCTCCAAAATAAG |
| Sequence-based reagent | *uex*$^{R622K}$, fragment 2, Forward | This study | PCR primers | GTACTTATTTTGGA GGGTAAAGTAGA AGTCACAATTGGC |
| Sequence-based reagent | *uex*$^{R622K}$, fragment 2, Reverse | This study | PCR primers | TTAGGGCTTACTT TGCTTGCTCTCGAATTTG |
| Sequence-based reagent | *CNNM2* cDNA, Forward | This study | PCR primers | ATCGCTCGAGATGA TTGGCTGTGGCGCTTGTG |
| Sequence-based reagent | *CNNM2* cDNA, Reverse | This study | PCR primers | ATCGTCTAGACTAT GCGTAGTCTGGCACGTCG |
| Sequence-based reagent | MagFRET-1 CDS, Forward | This study | PCR primers | ATCGCTCGAGGCCA CCATGGGCCATATGGTGAGC |
| Sequence-based reagent | MagFRET-1 CDS, Reverse | This study | PCR primers | ATCGTCTAGATTACTTG TACAGCTCGTCCATGCCGAG |
| Sequence-based reagent | MagIC CDS, Forward | This study | PCR primers | CACCAGGATGGCCAT CATCAAGGAGTTCATG |
| Sequence-based reagent | MagIC CDS, Reverse | This study | PCR primers | CCGTTACTCGATG TTGTGGCGGATCTTGAA |
| Sequence-based reagent | MARIO CDS, Forward | This study | PCR primers | CACCAGGGCTTGG TACCGAGCTCGGAT |
| Sequence-based reagent | MARIO CDS, Reverse | This study | PCR primers | CCGCCACTGTGCTG GATATCTGCAGAATTCTTA |
| Sequence-based reagent | Inverse PCR of *uex*$^{MI01943}$, Set 1, Forward | This study | PCR primers | ATGATAGTAAA TCACATTACG3 |
| Sequence-based reagent | Inverse PCR of *uex*$^{MI01943}$, Set 1, Reverse | This study | PCR primers | CAATAATTTAAT TAATTTCCC3 |
| Sequence-based reagent | Inverse PCR of *uex*$^{MI01943}$, Set 2, Forward | This study | PCR primers | CAAAAGCAACT AATGTAACGG |
| Sequence-based reagent | Inverse PCR of *uex*$^{MI01943}$, Set 2, Reverse | This study | PCR primers | TTGCTCTTCTTG AGATTAAGGTA |
| Sequence-based reagent | qPCR of *Nmdar1*, Forward | This study | PCR primers | ATCCCTCGACG TACAACATTGG |
| Sequence-based reagent | qPCR of *Nmdar1*, Reverse | This study | PCR primers | GAGGTGCTTTA TTGTGGTGCTAA |
| Sequence-based reagent | qPCR of *Nmdar2*, Forward | This study | PCR primers | ACTGCTGGG CAACCTGAG |
| Sequence-based reagent | qPCR of *Nmdar2*, Reverse | This study | PCR primers | GATTTCCGTCT TGTACGACCA |

*Continued on next page*

*Continued*

| Reagent type (species) or resource | Designation | Source or reference | Identifiers | Additional information |
|---|---|---|---|---|
| Sequence-based reagent | qPCR of *GluRIA*, Forward | This study | PCR primers | TTTTCTGGCC GGAATTTAGTT |
| Sequence-based reagent | qPCR of *GluRIA*, Reverse | This study | PCR primers | CCTGTTCGAAG ATTGCACCT |
| Sequence-based reagent | qPCR of *GluRIIA*, Forward | This study | PCR primers | AACCACCAGAT GTCCATCAATG |
| Sequence-based reagent | qPCR of *GluRIIA*, Reverse | This study | PCR primers | GAAGGTGCGC CACTCATAGT |
| Sequence-based reagent | qPCR of *Gapdh*, Forward | This study | PCR primers | CTTCTTCAGCG ACACCCATT |
| Sequence-based reagent | qPCR of *Gapdh*, Reverse | This study | PCR primers | ACCGAACTCG TTGTCGTACC |
| Sequence-based reagent | qPCR of *Tbp*, Forward | This study | PCR primers | ACAGGGGCAA AGAGTGAGG |
| Sequence-based reagent | qPCR of *Tbp*, Reverse | This study | PCR primers | CTTAAAGTCGAGG AACTTTGCAG |
| Sequence-based reagent | qPCR of *Ef1α100E*, Forward | This study | PCR primers | GCGTGGGTTT GTGATCAGTT |
| Sequence-based reagent | qPCR of *Ef1α100E*, Reverse | This study | PCR primers | GATCTTCTCCT TGCCCATCC |
| Sequence-based reagent | *uex*[MI01943] *Minos* excision, Forward | This study | PCR primers | GTGCCAGACCA CTGCACCATC |
| Sequence-based reagent | *uex*[MI01943] *Minos* excision, Reverse | This study | PCR primers | CCGTACCTATGTC GATTCCCACCTC |
| Sequence-based reagent | *uexΔ* lesion | This study | CRISPR gRNA1 | ACTTTCCAGTAC CTTAGCAC [TGG] |
| Sequence-based reagent | *uexΔ* lesion | This study | CRISPR gRNA2 | GTCACTCCTCGC GGTACCAC [TGG] |
| Sequence-based reagent | Verification of *uexΔ*, set 1, Forward | This study | PCR primers | AAGACATGG ATTGGCGATTG |
| Sequence-based reagent | Verification of *uexΔ*, set 1, Reverse | This study | PCR primers | AAGTCGCCATG TTGGATCG |
| Sequence-based reagent | Verification of *uexΔ*, set 2, Forward | This study | PCR primers | CTGGGCATGG ATGAGCTGTA |
| Sequence-based reagent | Verification of *uexΔ*, set 2, Reverse | This study | PCR primers | CTGGAGCGC AACAATTCTCT |
| Sequence-based reagent | *uex*[T626NRR] lesion | This study | CRISPR gRNA | GGTCGTGTAGAA GTCACAAT [TGG] |
| Sequence-based reagent | *uex*[T626NRR] lesion | This study | ssODN | GTCTTTATATTTTCA CTCAAGGAAAAGCTG TCGACTTTTTTGTA CTTATTTTGGAGGG TAAAGTAGAAGTCAC AATTGCCAAGGAAGCG CTTATGTTTGAAAGCG GGCCCTTTACTTATT |
| Sequence-based reagent | Screen for *uex*[T626NRR], set 1, Forward | This study | PCR primers | GGTTATTCTCGTAT TCCAGTGTACGATGG |
| Sequence-based reagent | Screen for *uex*[T626NRR], set 1, Reverse | This study | PCR primers | GAGATTCAGCATCT AGAGACAAAGACGCAG |

Continued

| Reagent type (species) or resource | Designation | Source or reference | Identifiers | Additional information |
|---|---|---|---|---|
| Sequence-based reagent | Screen for $uex^{T626NRR}$, set 2, Forward | This study | PCR primers | CGGTCGGGTTAGT TACTCTGGAAGATG |
| Sequence-based reagent | Screen for $uex^{T626NRR}$, set 2, Reverse | This study | PCR primers | CGCGTAAGCATTCA CACTAGCTGAGTAAC |
| Sequence-based reagent | Screen for $uex^{T626NRR}$, set 3, Forward | This study | PCR primers | GGCTACTTTCCAGT ACCTTAGCACTGG |
| Sequence-based reagent | Screen for $uex^{T626NRR}$, set 3, Reverse | This study | PCR primers | CGCGTAAGCATTCAC ACTAGCTGAGTAAC |
| Sequence-based reagent | Screen for $uex^{T626NRR}$, set 4, Forward | This study | PCR primers | CGGAGGTTACTCAA TCAAGACGTGTTTC |
| Sequence-based reagent | Screen for $uex^{T626NRR}$, set 4, Reverse | This study | PCR primers | CGCGTAAGCATTCAC ACTAGCTGAGTAAC |
| Commercial assay or kit | Direct-zol RNA MiniPrep | Cambridge Bioscience | R2050 | |
| Commercial assay or kit | SuperScript III First-Strand Synthesis SuperMix | Invitrogen | 18080400 | |
| Commercial assay or kit | LightCycler 480 SYBR Green I Master | Roche | 04707516001 | |
| Commercial assay or kit | pENTR/D-TOPO cloning kit | Invitrogen | K240020 | |
| Commercial assay or kit | Gateway LR ClonaseTM II Enzyme mix | Invitrogen | 11791020 | |
| Commercial assay or kit | NEBuilder HiFi DNA Assembly Master Mix | New England Biolabs | E2621S | |
| Commercial assay or kit | ExoSAP-IT PCR Product Cleanup Reagent | Thermo Fisher | 78201 | |
| Chemical compound, drug | $MgCl_2$ | Sigma-Aldrich | M1028 | |
| Chemical compound, drug | $MgSO_4$ | Sigma-Aldrich | M3409 | |
| Chemical compound, drug | $CaCl_2$ | Sigma-Aldrich | 21115 | |
| Chemical compound, drug | KCl | Sigma-Aldrich | 60142 | |
| Chemical compound, drug | EDTA | Sigma-Aldrich | 324504 | |
| Chemical compound, drug | Forskolin | Sigma-Aldrich | F6886 | |

*Continued*

| Reagent type (species) or resource | Designation | Source or reference | Identifiers | Additional information |
|---|---|---|---|---|
| Chemical compound, drug | 1,9-Dideoxyforskolin | Sigma-Aldrich | D3658 | |
| Chemical compound, drug | Magnesium Green | Invitrogen | M3733 | |
| Chemical compound, drug | Sucrose | Sigma-Aldrich | S0389 | |
| Chemical compound, drug | Mineral oil | Sigma-Aldrich | M5904 | |
| Chemical compound, drug | 3-Octanol | Sigma-Aldrich | 218405 | |
| Chemical compound, drug | 4-Methyl-Cyclohexanol | Sigma-Aldrich | 66360 | |
| Chemical compound, drug | Paraformaldehyde | Fisher Scientific | 15713 | |
| Chemical compound, drug | Phosphate buffered saline tablets | Fisher Scientific | 1282–1680 | |
| Chemical compound, drug | Triton X-100 | Sigma-Aldrich | T9284 | |
| Chemical compound, drug | Vectashield antifade mounting medium | Vector Laboratories | H1000 | |
| Chemical compound, drug | TRIzol RNA isolation reagents | Thermo Fisher | 15596018 | |
| Software, algorithm | Prism 6.0 | GraphPad | RRID:SCR_002798 | https://www.graphpad.com |
| Software, algorithm | SnapGene Viewer 4.1 | SnapGene | RRID:SCR_015052 | https://www.snapgene.com |
| Software, algorithm | Geneious R10.2 | Geneious | RRID:SCR_010519 | https://www.geneious.com |
| Software, algorithm | Fiji/ImageJ 1.4 | NIH | RRID:SCR_002285 | https://imagej.nih.gov |
| Software, algorithm | MATLAB R2017b | Mathworks | RRID:SCR_013499 | https://www.mathworks.com |
| Software, algorithm | Python 3.7 | Python Software Foundation | RRID:SCR_008394 | https://www.python.org |
| Software, algorithm | Visual Studio Code 1.42 | Microsoft | | https://code.visualstudio.com |
| Software, algorithm | Adobe Illustrator CC | Adobe Systems | RRID:SCR_010279 | https://www.adobe.com |
| Software, algorithm | InterPro | EMBL-EBI | RRID:SCR_005829 | https://www.ebi.ac.uk/interpro |
| Software, algorithm | Phyre[2] | Genome3D | | http://www.sbg.bio.ic.ac.uk/~phyre2 |

*Continued on next page*

*Continued*

| Reagent type (species) or resource | Designation | Source or reference | Identifiers | Additional information |
|---|---|---|---|---|
| Software, algorithm | TM-align | Zhang Lab | | https://zhanglab. ccmb.med.umich. edu/TM-align/ |
| Software, algorithm | Chimera 1.11 | UCSF | RRID:SCR_004097 | https://www.cgl. ucsf.edu/chimera/ |

## Contact for reagent and resource sharing

A full list of reagents can be viewed in the Key Resources Table.

Further information and requests for resources and reagents should be directed to and will be fulfilled by the Lead Contact, Scott Waddell (scott.waddell@cncb.ox.ac.uk).

## Experimental model and subject details

### Fly strains

Unless stated otherwise, flies were raised on standard cornmeal food under a 12 hr light–dark cycle at 60% humidity and 25°C. Test and control flies for GAL80$^{ts}$ experiments were raised at 18°C. Mixed sex flies 1–7-days-old were used in experiments.

Canton-S was the wild-type strain. The GAL4 driver lines used in this study are c739-GAL4 (*McGuire et al., 2001*), c305a-GAL4 (*Krashes et al., 2007*), NP7175-GAL4 (*Tanaka et al., 2004*), 0770-GAL4 (*Gohl et al., 2011*), MB247-GAL4 (*Zars et al., 2000*), nSyb-GAL4 (Bloomington *Drosophila* Stock Centre, BDSC 51635), elav-GAL4 (BDSC, 8765), and uex-GAL4 (*Kvon et al., 2014*); Vienna *Drosophila* Resource Center, VDRC, VT23256-GAL4). The UAS lines obtained from the stock center are UAS-*CD8::GFP* (BDSC, 5136), UAS-*Nmdar*$^{RNAi}$ (BDSC, 25941), and UAS-*uex*$^{RNAi}$ (BDSC, 36116). The various mutant and transgenic lines are described, uex$^{MI01943}$ (*Venken et al., 2011*; BDSC, 32805), uex$^{NC1}$ (BDSC, 7167), rut$^{2080}$ (*Han et al., 1992*), and dnc$^{1}$ (*Dudai et al., 1976*), tubP-GAL80$^{ts}$ (*McGuire et al., 2003*) and PhsILMiT (BDSC, 24613). The uex$^{MI01943.ex1}$ and uex$^{MI01943.ex2}$ *Minos* excision lines were generated using the procedure described in *Arcà et al., 1997*. The detailed mating scheme is shown in *Figure 2—figure supplement 2A*. Potential excision lines were established from individual flies exhibiting the *yellow* body color phenotype. Genomic DNA was extracted from six such lines and DNA flanking the uex$^{MI01943}$ MiMIC was amplified by PCR and sequenced. The uex$^{MI01943.ex1}$ and uex$^{MI01943.ex2}$ lines were identified to harbor precise excisions, having restored the wild-type genomic sequence. See Resource Table for PCR and sequencing primer sequences. Schematic of the sequence detail of the uex$^{MI01943}$ MiMIC insertion and in the excisions is shown in *Figure 2—figure supplement 2B*. To construct UAS-*uex* transgenic flies a full-length *uex* coding sequence (CDS) was cloned by RT-PCR. Total RNA was isolated from wild-type flies using TRIZOL (Thermo Fisher, 15596018) and reverse transcribed into cDNA using SuperScript III first-strand synthesis system (Invitrogen, 18080400). This total cDNA mix was used as a template to amplify the *uex* CDS. See Resource Table for primer sequences. The PCR product was digested with *SacII* and *XhoI* and then ligated into the complementary sites of pUAST (*Brand and Perrimon, 1993*). The pUAST cloned *uex* CDS was fully sequenced and verified to represent the 2505 bp of the wild-type *uex* cDNA reading frame (note, all four possible *uex* mRNA isoforms, FlyBase Release 6, encode the same 834 amino acid protein). UAS-*uex* transgenic flies were generated commercially (Bestgene) by transformation with the pUAST-*uex* vector. We mapped the UAS-*uex* chromosome insertion of 10 independent transgenic lines and behaviorally tested three lines, denoted UAS-*uex*$^{3M}$, UAS-*uex*$^{5M}$ and UAS-*uex*$^{8M,}$ with an insert on the third chromosome. UAS-*uex*$^{3M}$ flies were those used throughout the study and referred to as UAS-*uex* in the manuscript.

UAS-*uex*$^{R622K}$ transgenic flies were generated similar to UAS-*uex* flies. A missense mutation was introduced at codon 622 of UEX within the CNBH domain, mimicking that previously engineered in the cAMP-binding domain of the regulatory subunit of protein kinase A (*Bubis et al., 1988*). The mutation changes the CGT codon encoding Arg into AAA encoding Lys. The mutation was introduced into the wild-type *uex* CDS using Gibson Assembly Master Mix (New England Biolabs, E2621S) as described in 'Improved methods for site-directed mutagenesis using Gibson Assembly

Master Mix' (NEB Application Note). The primer sets used are detailed in the Resource Table. The product of Gibson assembly was further amplified by PCR and the resulting product was cloned into the pUAST vector and sequenced. Transgene insertions were mapped as for UAS-*uex* and one of two insertions mapped to the third chromosome was used in behavior experiments.

UAS-*CNNM2*, UAS-*CNNM2*[E122K], UAS-*CNNM2*[E357K], UAS-*CNNM2*[S269W], and UAS-*CNNM2*[T568I] transgenic fly lines were generated by transformation with pUAST constructs containing wild-type or point mutated versions of a mouse *CNNM2* cDNA tagged with HA (mCNNM2::HA), described in *Arjona et al., 2014*. Wild-type or mutated versions of *CNNM2* were amplified from original mCNNM2::HA clones in pCiNEO_IRES_GFP plasmids (*Arjona et al., 2014*). Primers are detailed in the Resource Table. PCR products were digested with *XhoI* and *XbaI* and ligated into the complementary sites in pUAST. Insertions of each construct on the third chromosome were identified by mapping as described above and were used in the behavior experiments. Note that all *CNNM2* encoding constructs used in the study are HA tagged, although the notation is often omitted for brevity.

UAS-*MagFRET-1* transgenic fly lines were generated by transformation with pJFRC-MUH constructs containing MagFRET-1 CDS, which was sub-cloned from the pCMVMagFRET-1 plasmid, described in *Lindenburg et al., 2013*. Primers are detailed in the Resource Table. PCR products were digested with *XhoI* and *XbaI* and ligated into the complementary sites in pJFRC-MUH. Insertion of the construct was mediated by the site-specific transgenesis system and the landing site is attP2 (on the third chromosome).

UAS-*MagIC* and UAS-*MARIO* transgenic fly lines were generated by transformation with pTW constructs containing the MagIC/MARIO CDS, which were sub-cloned from the plasmids MagIC/pcDNA3 and MARIO/pcDNA3, kindly provided by T. Nagai: (*Maeshima et al., 2018* and *Koldenkova et al., 2015*). MagIC/MARIO CDS were first PCR amplified from MARIO/pcDNA3 and MagIC/pcDNA3 respectively and were cloned into the pENTR/D-TOPO vector. Primers are detailed in the Resource Table. Note that the MARIO sense primer was designed to overlap with the sequence of pcDNA3 at the insertion site of MARIO. MagIC/MARIO CDS were further cloned into the Gateway destination vector pTW (*Drosophila* Gateway Vector Collection).

The CRISPR/Cas9 edited *uexΔ* locus was generated commercially by GenetiVision. The editing scheme is shown in *Figure 2—figure supplement 2C*. The *uex* locus sits in reverse orientation on chromosome 2R, spanning a 49,141 bp region between position 3,900,285 and 3,949,425 (FlyBase, Release 6). The following description relates to these coordinates within the *uex* locus. To generate *uexΔ*, two gRNA plasmids and one double strand DNA donor (dsDNA) plasmid were constructed and injected into *nos*-Cas9 embryos (BDSC, 54591). As indicated in *Figure 2—figure supplement 2C* and detailed in the Resource Table, the upstream gRNA1 lies in Exon 6 and targets sequence 30,930.30,952. The corresponding downstream gRNA2 lies between Exon 7 and Exon 8 between 33,988 and 34,010. Both gRNAs were individually cloned into pCFD3-dU63gRNA (Addgene, 49410). The cut site of gRNA1 should be between 30,946 and 30,947 while gRNA2 should lead to a cut between 33,993 and 33,994. A 795 bp upstream homology arm (30,152.30,946) and 977 bp downstream homology arm (33,994.34,970) were cloned into the donor DNA plasmid. A termination codon (STOP, in all three reading frames) was inserted between the two homology arms and followed by a GFP cassette driven by a 3xP3 promoter. The donor DNA backbone was engineered by GenetiVision and the complete donor sequence for the *uexΔ* line is available upon request. Successful editing was identified by expression of GFP in the fly eyes and confirmed by genomic PCR and sequencing. In the *uexΔ* flies, a 3047 bp fragment from 30,947 to 33,993 was replaced by the sequence between the two homology arms in the donor plasmid, mainly the STOP signal and GFP cassette. The *uexΔ* allele truncates the *uex* ORF. Primers used for genomic PCR verification are detailed in the Resources Table. The *nos*-Cas9 transgene (on X chromosome) was removed by crossing.

CRISPR/Cas9-edited *uex::HA* flies were generated by WellGenetics using the ScarlessDsRed system developed by Kate O'Connor-Giles' lab (unpublished, original plasmid donated to Addgene, #80822). A 6XHA tag was fused in frame to the carboxy-terminus of UEX by inserting the 6XHA-coding sequence immediately prior to the native STOP codon in the *uex* locus (*Figure 3—figure supplement 1A*). The process involved two main steps. In step 1, a 6XHA tag together with a pBAC transposon containing a DsRed cassette were inserted in frame with the STOP codon of *uex* using CRISPR/Cas9-mediated genome editing by homology-directed repair (HDR) using 1 gRNA and one

dsDNA plasmid donor. The gRNA lies −50 bp from the *uex* STOP codon and should direct a cut between 48,587 and 48,588. The gRNA was cloned into a pCFD3-dU63gRNA plasmid. A 1,200 bp upstream arm (47,438.48,637) and 1,033 bp downstream arm (48,641.49,673) were cloned into the donor DNA plasmid with the pUC57-Kan (2579 bp) backbone. See Resource Table for gRNA and primer sequences. A Protospacer Adjacent Motif (PAM) mutation (TCC to TCG, 48,581.48,583) was introduced in the donor to promote HDR. A 6XHA tag, followed by a pBAC transposon containing a 3XP3 promoter-driven DsRed cassette, was inserted between the two homology arms. A pBAC recognition motif TTAA is embedded in the STOP codon of 6XHA. The complete donor sequence is available upon request. Donor and gRNA plasmids were injected into *nos-Cas9* embryos (NIG-FLY, CAS0002). Successful editing was identified by expression of DsRed in the fly eyes and confirmed by genomic PCR and sequencing. Six independent positive lines were identified and four passed PCR validation. Of these four lines, one further passed sequencing validation and is the intermediate line represented in *Figure 3—figure supplement 1A*. Four isogenized and balanced stocks were established from this line. In step 2, the DsRed selection marker was excised by *PiggyBac (PBac)* transposition with the helper line *Tub-PBac* (BDSC, 8285). Five homozygous viable lines with successful excision were validated by genomic PCR and sequencing. One designated *uex::HA* was used in experiments in the manuscript.

To construct the CRISPR/Cas9-edited *uex*^T626NRR flies, we designed and cloned a gRNA and designed and ordered (Sigma) a single-stranded oligo-deoxynucleotide (ssODN). gRNA and ssODN sequences are detailed in the Resource Table. As we planned to make a single amino acid substitution R622K in the UEX CNBH domain, the 120 bp ssODN donor was centered on codon R622 and carries the codon change CGT to AAA (at 31,179.31,181) corresponding to R622K. The expected cut site of the gRNA (between 31,192 and 31,193) is only 11 bp away from the expected mutation point. To enhance the likelihood of HDR, which is reportedly low using ssODN as donor, we commercially (GenetiVision) injected editing material into 250 *lig4* KO *vasa*-Cas embryos (*Zimmer et al., 2016*). We obtained 37 viable G0 flies from the injected embryos. A total of 224 G1 flies were subjected to single fly genomic PCR and sequencing to screen for the expected mutation. Primers detailed in Resource Table. We identified 59 putative edited lines from first-round screening, and of these 12 were confirmed. Despite using *lig4* KO *vasa*-Cas9, we detected only non-homologous end joining (NHEJ) events instead of HDR-mediated point mutations. Of the 12 edited lines, six were homozygous lethal and the other six were viable. In four of the homozygous viable lines, we found a replacement of G with T at position 31,192 together with a 6 bp in frame insertion of ATCTTC between 31,192 and 31,193. This NHEJ editing corresponds to the T626 → NRR change in the protein sequence of UEX (*Figure 5A*). The X chromosome *vasa*-Cas9 was removed from these lines by crossing and one line referred to as *uex*^T626NRR was used in the behavior experiments in the manuscript.

## Method details

### Behavioral experiments

For behavioral T-maze experiments, 1–7-day-old mixed sex flies were used. Odors were 4-methylcyclohexanol (MCH) and 3-octanol (OCT), diluted ~1:10$^3$ (specifically, 9 μl MCH or 7 μl OCT in 8 ml mineral oil). All experiments were performed at 23°C and 55–65% relative humidity.

Appetitive immediate and later memory experiments were performed essentially as described (*Krashes and Waddell, 2008*; *Perisse et al., 2013*). Batches of 100–120 flies were starved for 21–23 hr before training in 35 ml starvation vials containing ~2 ml 1% agar (as a water source) and a 2 cm × 4 cm filter paper. Sugar papers (5 cm × 7.5 cm) for training were prepared by soaking with 4 ml of 2 M sucrose and drying overnight. Water papers of same size were soaked with water and left overnight. For appetitive training, flies were transferred from a starvation tube to a training tube with a dry 'water' paper, and immediately attached to the training arm of the T-maze and exposed to the CS− odor for 2 min, followed by 30 s of clean air. Flies were then transferred to another training tube with dry sugar paper, attached to the T-maze and exposed to the CS+ odor for 2 min. Immediate memory was tested by transporting flies to the T-choice point and allowing them 2 min to choose between the two odor streams. To assay 24 hr memory, flies were removed from the training tube and transferred to standard cornmeal food vials for 1 hr, then transferred back into starvation vials for 23 hr until testing. Performance Index was calculated as the number of flies in the CS+

arm minus the number in the CS− arm, divided by the total number of flies. MCH and OCT were alternately used as CS+ or CS− and a single sample, or n, represents the average Performance Index from two reciprocally trained groups.

For behavior tests after $Mg^{2+}$ feeding, 1–2-day-old flies were housed in vials with $Mg^{2+}$ supplemented food for 1–5 days before being starved for appetitive training and testing, as described above. To make 80 mM $[Mg^{2+}]$ food, 40 ml of 1 M $MgCl_2$ solution was added to 460 ml of normal liquid fly food; 1 mM $[Mg^{2+}]$ food was made by diluting 0.5 ml 1 M $MgCl_2$ in 39.5 ml MilliQ water and adding it to 460 ml liquid food. Food was aliquoted and cooled to solidify. $MgSO_4$ and $CaCl_2$ supplemented food was prepared the same way.

Aversive immediate and 24 hr memory experiments were conducted as previously described (*Hirano et al., 2013*; *Perisse et al., 2016*; *Tully and Quinn, 1985*). Groups of 100–120 flies were trained with either one cycle of aversive training, or five cycles spaced by 15 min inter-trial intervals (spaced training). For aversive immediate memory, flies were tested after one-cycle training. Aversive 24 hr memory was tested using two different protocols. In the fasting-facilitated protocol, flies were starved for 16 hr before one-cycle training (*Hirano et al., 2013*). For spaced training, flies were not starved before training. Flies were fed on normal fly food for 24 hr after fasting-facilitated and spaced training, before being tested for memory performance. During each aversive training cycle, flies were exposed for 1 min to a first odor (CS+) paired with twelve 90 V electric shocks at 5 s intervals. Following 45 s of clean air, a second odor (CS−) was presented for 1 min without shock. Performance Index was calculated as the number of flies in the CS− arm minus the number in the CS+ arm, divided by the total number of flies. MCH and OCT were alternately used as CS+ or CS− and a single sample, or n, represents the average Performance Index from two reciprocally trained groups.

Sensory acuity tests (*Figure 2—source data 1*) were performed as described (*Keene et al., 2004*; *Keene et al., 2006*; *Schwaerzel et al., 2003*) with modifications. To test olfactory acuity, untrained flies were given 2 min to choose between a diluted odor as used in conditioning and air bubbled through mineral oil in the T maze. An Avoidance Index was calculated as the number of flies in the air arm minus the number in the odor arm, divided by the total number of flies. Electric shock avoidance was performed and calculated similarly. Untrained flies chose for 1 min between two tubes containing electric grids, but only one was connected to the power source. An avoidance index was calculated as the number of flies in the non-electrified arm minus the number in the electrified arm, divided by the total number of flies. To assess sugar acuity, starved flies were given 2 min to choose between an arm of the T-maze containing a dried sugar paper and the other containing a dried 'water' filter paper. Both papers were prepared as in the appetitive memory assays. A Preference Index was calculated as the number of flies in the sugar arm minus that in the other arm, divided by the total number of flies. We found that keeping the light on in the behavioral room and having air flow running through the testing tubes greatly enhanced the Preference Index in wild-type flies and therefore applied those conditions for all sugar preference testing.

## Anti-UEX antibody and western blot

A polyclonal UEX antibody was developed commercially by Eurogentec. Two peptides were synthesized as antigens: Peptide 1 H-CLPKLDDKFESKQSKP-OH (16aa) and Peptide 2 H-CVDNRTK TRRNRYKKA-NH2 (16aa) and injected into rabbits. Only Peptide 2 induced a robust immune response and was processed further. The final serum was purified against Peptide 2 and used for western blot analysis as a 1:2000 dilution.

For each sample in western blot, proteins were extracted from 20 fly heads by homogenizing thoroughly in 120 µl of protein sample buffer containing a mixture of 30 µl 2-mercaptoethanol (Bio-Rad), 270 µl 4× Laemmli sample buffer (BioRad), and 900 µl Nuclease Free Water (Invitrogen). Samples were then boiled on a 100℃ heat block for 3 min and centrifuged for 10 min before loading. A sample volume equivalent to four heads was loaded into each SDS-PAGE gel lane. Proteins were transferred to PVDF membrane and blocked in 5% skim milk for 1 hr at 25℃ with 35 rpm agitation. Membrane was then incubated in anti-UEX solution (1:2000 rabbit anti-UEX in 5% skim milk) overnight at 4℃ with 35 rpm agitation. Membrane was washed quickly three times followed by 3 × 10 min washes in TBST solution (100 ml of TBS 10× solution, BioRad, diluted in 900 ml of MilliQ water, with 0.1% Tween 20) and then incubated with HRP-conjugated secondary antibody solution (1:5000 of goat anti-rabbit in 5% skim milk) for 1–2 hr at 25℃ with 35 rpm agitation. The membrane was again washed quickly for three times followed by 3 × 10 min washes in TBST. Protein bands were

visualized using Pierce ECL western blotting substrate (Life technologies, 32134). Membrane was then stripped using Millipore ReBlot Plus Mild solution (Merck, 2502), blocked again in 5% skim milk, and probed with mouse anti-Tubulin primary antibody (1:2000, Sigma, T6199) and corresponding HRP conjugated goat anti-mouse secondary antibody (1:5000) following the protocol detailed above.

## Immunostaining

Immunostaining was performed as described (*Wu and Luo, 2006*). Brains from 1- to 5-day-old adult flies were dissected in PBS and fixed for 20 min in PBS with 4% paraformaldehyde at room temperature. They were then washed twice briefly in 0.5% PBT (2.5 ml Triton-X100 in 497.5 ml PBS) and three 20 min washes. Brains were then blocked for 30 min at room temperature in PBT containing 5% normal goat serum and then incubated with primary and secondary antibodies with mild rotation (35 rpm) at 4°C for 1 or 2 days. Primary antibodies were rabbit anti-*GFP* (1:250; Invitrogen A11122) and rabbit anti-HA (1:250, NEB 3724T). Alexa 488–conjugated goat anti-rabbit (1:250; Invitrogen, A11034) was the secondary antibody. Before and after the secondary antibody incubation, brains were subjected to two quick washes followed by three 20 min washes in 0.5% PBT. Stained brains were mounted on glass slides in Vectashield (Vector Labs H1000) and imaged using a Leica TCS SP5 confocal microscope at 40× magnification (HCX PL APO 40×, 1.3 CS oil immersion objective, Leica). Image stacks were collected at 1024 × 1024 resolution with 1 μm steps and processed using Fiji (*Schindelin et al., 2012*). For quantification in *Figure 3G and H*, rectangular ROIs of approximately 40 × 25 μm for the for γ lobe, or round ROIs with diameter of 15 μm for αβ, α′β′, and EB were manually drawn on a single section of a z-stack scan of the fly brain. Corresponding ROIs were also drawn on the superior medial protocerebrum (SMP) as a background control region, and the mean fluorescence was calculated using ImageJ. ROI intensity of the MB lobes and the EB was normalized to that of the respective SMP intensity. An average between left and right brains was used for a single data point. For quantification in *Figure 7C and D*, ROIs are indicated in the figures and ROI intensity was calculated similar to results in *Figure 3H*. In *Figure 7C*, a line was drawn through the widest part of the tip of the α lobe. The intensity profile of this line was obtained through ImageJ. Thirty data points in the middle of such a profile spanning about a 15 μm line were extracted for each line profile. The profile was further normalized to the mean value of the first five data points ($F_0$) and calculated as $(F-F_0)/F_0$. Mean values of these normalized profiles from different brains were plotted (*Figure 7C*, middle panel). Left and right profiles of brains were calculated and are separately displayed. In *Figure 7D*, the relative intensities from different ROIs representing different regions are added together to generate a total intensity measure for the MB.

The human *CNNM4* cDNA expression construct used to investigate $Mg^{2+}$ efflux in cell culture is that described previously (*Yamazaki et al., 2013*). A construct expressing *Drosophila uex* was generated by inserting a FLAG tag in front of the STOP codon of the *uex* CDS. FLAG-tagged *CNNM4* and *uex* cDNAs were subsequently inserted into pCMV tag-4A (Agilent) for expression in HEK293 cells. HEK293 cells were cultured in Dulbecco's modified Eagle medium (Nissui) supplemented with 10% Fetal Bovine Serum (FBS) and antibiotics. Expression plasmids were transfected with Lipofectamine 2000 (Invitrogen).

For immunostaining, cells were fixed with 3.7% formaldehyde in PBS for 20 min and then permeabilized with 0.2% Triton X-100 in PBS for 5 min, both at room temperature. They were next blocked with PBS containing 3% FBS and 10% bovine serum albumin (blocking buffer) for 1 hr at room temperature. Cells were then incubated overnight at 4°C with rabbit anti-FLAG antibody (F7425, Sigma-Aldrich) diluted in blocking buffer, washed 3× with PBS, and incubated for 1 hr at room temperature with Alexa 488-conjugated anti-rabbit IgG (Invitrogen) and rhodamine-phalloidin (for F-actin visualization, Invitrogen) diluted in blocking buffer. After three washes with PBS, coverslips were mounted on slides and imaged with a confocal microscope (FluoView FV1000; Olympus).

$Mg^{2+}$-imaging with Magnesium Green was performed as described (*Yamazaki et al., 2013*), with slight modifications. To avoid potentially decreasing $[Mg^{2+}]_i$ with the expressed proteins, transfected HEK293 cells were cultured in growth media supplemented with 40 mM $MgCl_2$ until imaging. Cells were then incubated with $Mg^{2+}$-loading buffer (78.1 mM NaCl, 5.4 mM KCl, 1.8 mM $CaCl_2$, 40 mM $MgCl_2$, 5.5 mM glucose, and 5.5 mM HEPES-KOH [pH 7.4]), including 2 μM Magnesium Green-AM (Invitrogen), for 30 min at 37°C. Cells were then rinsed once with loading buffer and viewed with an Olympus IX81 microscope equipped with an ORCA-Flash 4.0 CMOS camera (Hamamatsu) and a

SHI-1300L mercury lamp (Olympus). Fluorescence was measured every 20 s (excitation at 470–490 nm and emission at 505–545 nm) under the control of Metamorph software (Molecular Devices). Buffer was then changed to $Mg^{2+}$free buffer ($MgCl_2$ in the loading buffer was replaced with 60 mM NaCl). Data are presented as line plots (mean of 10 cells). After imaging, cells were fixed with PBS containing 3.7% formaldehyde and subjected to immunofluorescence microscopy to confirm protein expression.

## FRET-based $Mg^{2+}$ concentration measurements in fixed fly brains

One- to two-day-old flies with genotype c739; UAS-*MagFRET-1* were housed in vials with 1 mM or 80 mM [$Mg^{2+}$] food for 4 days before being collected. Fly brains were dissected in PBS and fixed for 20 min in PBS with 4% paraformaldehyde at room temperature. They were then washed twice briefly in 0.5% PBT (2.5 ml Triton-X100 in 497.5 ml PBS) and three 10 min washes. Brains were then mounted on glass slides in Vectashield (Vector Labs H1000) and imaged using a wide-field Scientifica Slicescope with a 40×, 0.8 NA water-immersion objective and an Andor Zyla sCMOS camera with Andor Solis software (v4.27). In order to get the FRET ratio that indicates the $Mg^{2+}$ concentration of the αβ neuron, time series were acquired alternatively between the cerulean channel and the citrine channel at 3 Hz with 512 × 512 pixels and 16 bit. The excitation wavelength for both channels is 436 nm, while the emission filter for cerulean is 460–500 nm and that for citrine is 520–550 nm. Series acquisition starts from the cerulean channel and lasts for 5 s, then switches to the citrine channel and last for another 5 s, and this cycle is repeated for two more times. A total of 30 s (90 frames) image stack was therefore acquired for each brain. Image stacks were subsequently analyzed using ImageJ and custom-written Matlab scripts. In brief, rectangle ROIs (*Figure 1E*, left panel) were manually drawn on the αβ lobes (one on α lobe and one on β lobe for each hemisphere), and outside the αβ lobes (one for each hemisphere) as background control. Fluorescence intensity from the cerulean channel was calculated by dividing each vertical or horizontal lobe ROI by the background ROI, and averaged between the two hemispheres for each lobe, and averaged over the 15 frames for each cycle. That from the citrine channel was obtained similarly. A FRET ratio was obtained from the above intensities, further averaged among the three cycles of acquisition, depicted as one data point in *Figure 1E* (right panel).

## Confocal $Mg^{2+}$ imaging in explant fly brain

Explant brains expressing *c739*-GAL4 driven UAS-*MagIC* were placed at the bottom of a 35 mm glass bottom microwell dish (Part No. P35G-1.5–14 C, MatTek Corporation), beneath extracellular saline buffer solution (103 mM NaCl, 3 mM KCl, 5 mM N-Tris, 10 mM trehalose, 10 mM glucose, 7 mM sucrose, 26 mM $NaHCO_3$, 1 mM $NaH_2PO_4$, 1.5 mM $CaCl_2$, 4 mM $MgCl_2$, osmolarity 275 mOsm [pH 7.3]) following dissection in calcium-free buffer (*Barnstedt et al., 2016*). To determine the $Mg^{2+}$ sensitivity of UAS-*MagIC* as well as the response of UAS-*MagIC* to other chemicals such as EDTA, EGTA, and $CaCl_2$ (*Figure 8B*), brains were incubated in the saline buffer solution with 20 µg/ml digitonin for 6 min before imaging (*Koldenkova et al., 2015*). To investigate the $Mg^{2+}$ fluctuation in response to Forskolin (FSK) application (*Figure 8C–I*), brains were put in the saline buffer solution without digitonin or incubation. In both situations, saline refers to the buffer (either with or without digitonin) in which the brain is submerged.

Imaging was carried out in a LSM780 confocal microscope (Zeiss) with a 20× air objective using the ZEN 2011 software. The Venus part of MagIC was excited with a 488 nm laser and its emission was collected in the 520–560 nm range. mCherry was excited with a 561 nm laser and its emission was collected in the 600–640 nm range. Time series were acquired at 0.5 Hz with 512 × 512 pixels and 16 bit. Following 60 s of baseline Venus/mCherry measurement, 2–20 µl of saline or other relevant chemical solution was added via a micropipette to the dish with constant image capture. The effects of applied agents on Venus/mCherry emission were then recorded for 15–20 min.

Image stacks were subsequently analyzed using ImageJ and custom-written Python scripts. In brief, rectangle ROIs were manually drawn on the αβ neurons (one for each hemisphere, *Figure 8A*), and another ROI of the same size was drawn in the middle but outside the MBs as background control. Fluorescence intensity from the Venus (or mCherry) channel was calculated by subtracting the background ROI from the calyx ROIs, respectively, and averaged between the two hemispheres. This is referred as 'Rel. Intensity (a.u.)' in *Figure 8D and E*. The ratio between Venus and mCherry

intensity was calculated as 'MagIC Ratio' in *Figure 8B and C* and *Figure 8F and G*. For *Figure 8H*, the intensity for the two channels was calculated separately. In this case, 'Rel. Intensity ($\Delta F/F_0$)' refers to the relative fluorescence intensity normalized to the mean intensity from the baseline period $F_0$, calculated as $(F-F_0)/F_0$. The relative intensity $\Delta F/F_0$ of Venus was used to calculate the PSD (*Figure 8I*) through python function psd (under matplotlib.pyplot), which adopted a Welch's average periodogram method (*Bendat et al., 2000*).

### Reverse transcription and quantitative real-time PCR

For each sample, 120 flies were frozen in liquid nitrogen and their heads were homogenized completely in TRIzol reagent (Invitrogen). Total RNA was extracted using Direct-zol RNA MiniPrep (R2050) kit following the manufacturer's instructions. cDNA was synthesized using SuperScript III First-Strand synthesis System (Invitrogen). Five independent samples were prepared for each different treatment or genotype. Quantitative PCR was performed in triplicate for each cDNA sample on a LightCycler 480 Instrument (Roche) using SYBR Green I Master Mix (Roche). Melting curves were analyzed after amplification, and amplicons were visualized by agarose gel electrophoresis to confirm primer specificity. Relative transcript levels were calculated by the $2^{-\Delta\Delta Ct}$ method (*Livak and Schmittgen, 2001*), and the geometric mean of the $C_t$ values of three reference genes (*Gapdh*, *Tbp*, and *Ef1α100E*) was used for normalization. Primers are detailed in the Resource Table.

### Inverse PCR

Inverse PCR was used to map the MiMIC insertion position in *uex*[MI01943] flies. Genomic DNA was prepared from 15 adult flies. DNA equivalent to two flies was then digested in a 25 µl restriction reaction with *Mbo I* and 10 µl of the product was ligated overnight at 4°C overnight to circularize the fragments; 5 µl of the ligation product was used for inverse PCR. PCR product was purified using Exo/SAP reaction (Thermo Fisher, 78201) before being sequenced. Sequence was compared to the *D. melanogaster* genome (FlyBase, Release 6) by BLAST and matched uniformly to the region 3,882,886.3,882,641 on 2R, consistent with the reported *uex*[MI01943] insertion on FlyBase. Primers detailed in the Resource Table.

### Protein domain prediction and alignment

Protein sequence alignment was carried out using Geneious R10.2.2. Protein domain prediction was performed with InterPro (*Finn et al., 2017*; *Jones et al., 2014*) and Phyre[2] (*Kelley et al., 2015*). Protein domain and structure alignment was performed using TM-align (*Zhang and Skolnick, 2005*). Protein structure visualization was rendered in Chimera 1.11.2 (*Pettersen et al., 2004*).

### Quantification and statistical analyses

Behavior data were analyzed using Excel and Prism 6. Imaging data were analyzed using ImageJ and custom-written MATLAB or Python scripts. Unpaired two-tailed t-tests were used for comparing two groups, and one-way ANOVA followed by a Tukey's post-hoc test was used for comparing multiple groups. Threshold of statistical significance was set at $p < 0.05$.

## Acknowledgements

We thank F J Arjona and J G J Hoenderop for the murine CNNM2 clones and comments on the manuscript. We are grateful to T Nagai for clones of MagIC and MARIO and to the Bloomington Stock Center and VDRC for flies. We thank P Cognigni, Y Huang, R Brain, R Szoke-Kovacs, and M Goodwin for technical support and other members of the Waddell group for discussion. We acknowledge N Halidi, C Monico, and the Micron Advanced Bioimaging Unit (supported by Wellcome Strategic Awards 091911/B/10/Z and 107457/Z/15/Z) for their support and assistance in this work. E M was funded by an EMBO Long-term fellowship (ALTF 184-2109). K D J acknowledges support from the Rhodes Trust. S W was funded by a Wellcome Principal Research Fellowship (200846/Z/16/Z) and an ERC Advanced Grant (789274).

## Additional information

### Funding

| Funder | Grant reference number | Author |
|---|---|---|
| Wellcome | 200846/Z/16/Z | Scott Waddell |
| European Commission | 789274 | Scott Waddell |
| EMBO | ALTF 184-2019 | Eleonora Meschi |

The funders had no role in study design, data collection and interpretation, or the decision to submit the work for publication.

### Author contributions

Yanying Wu, Conceptualization, Formal analysis, Validation, Investigation, Visualization, Methodology, Writing - original draft, Writing - review and editing; Yosuke Funato, Formal analysis, Validation, Investigation, Visualization, Methodology; Eleonora Meschi, Formal analysis, Investigation; Kristijan D Jovanoski, Formal analysis, Investigation, Methodology; Hiroaki Miki, Resources, Supervision, Funding acquisition, Methodology; Scott Waddell, Conceptualization, Resources, Supervision, Funding acquisition, Methodology, Writing - original draft, Project administration, Writing - review and editing

### Author ORCIDs

Eleonora Meschi (iD) http://orcid.org/0000-0003-2401-7969
Scott Waddell (iD) https://orcid.org/0000-0003-4503-6229

### Decision letter and Author response

Decision letter https://doi.org/10.7554/eLife.61339.sa1
Author response https://doi.org/10.7554/eLife.61339.sa2

## Additional files

### Supplementary files

• Transparent reporting form

### Data availability

Behaviour data from T-maze assays are deposited in Dryad Digital Repository (https://doi.org/10.5061/dryad.q2bvq83hs). All other data generated or analysed during this study are included in the manuscript and supporting files.

The following datasets were generated:

| Author(s) | Year | Dataset title | Dataset URL | Database and Identifier |
|---|---|---|---|---|
| Wu Y, Funato Y, Meschi E, Jovanoski KD, Miki H, Waddell S | 2020 | Behavior data from T-maze assay | http://dx.doi.org/10.5061/dryad.q2bvq83hs | Dryad Digital Repository, 10.5061/dryad.q2bvq83hs |
| Wu Y, Funato Y, Meschi E, Jovanoski KD, Miki H, Waddell S | 2020 | Imaging data from ex-vivo MagIC assay Part II | http://dx.doi.org/10.5061/dryad.zpc866t7d | Dryad Digital Repository, 10.5061/dryad.zpc866t7d |
| Wu Y, Funato Y, Meschi E, Jovanoski KD, Miki H, Waddell S | 2020 | MagFRET signal from fixed brain | http://dx.doi.org/10.5061/dryad.dv41ns1wp | Dryad Digital Repository, 10.5061/dryad.dv41ns1wp |
| Wu Y, Funato Y, Meschi E, Jovanoski KD, Miki H, | 2020 | Imaging data from ex-vivo MagIC assay Part I | http://dx.doi.org/10.5061/dryad.k0p2ngf6z | Dryad Digital Repository, 10.5061/dryad.k0p2ngf6z |

Waddell S

| Wu Y, Funato Y, Meschi E, Jovanoski KD, Miki H, Waddell S | 2020 | Immuno-Fluorescence data from confocal scanning | http://dx.doi.org/10.5061/dryad.80gb5mkpx | Dryad Digital Repository, 10.5061/dryad.80gb5mkpx |

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
