## [Decision Letter]

**Acceptance summary:**

Wu et al. discover elevated dietary magnesium enhances long-term memory in *Drosophila*. They then identify target neurons (α/β Kenyon Cells of the mushroom body) and a protein critical for the effect, the CNNM2 ortholog Unextended (UEX), which the authors implicate in magnesium efflux. The study is well-performed and conclusions justified by the data presented. Overall, this paper marks an important step in examining a highly significant, but understudied topic.

**Decision letter after peer review:**

Thank you for submitting your article "Magnesium efflux from *Drosophila* Kenyon Cells is critical for normal and diet-enhanced long-term memory" for consideration by *eLife*. Your article has been reviewed by two peer reviewers, and the evaluation has been overseen by K VijayRaghavan as the Senior Editor and Reviewing Editor. The reviewers have opted to remain anonymous.

The reviewers have discussed the reviews with one another and the Reviewing Editor has drafted this decision to help you prepare a revised submission.

Summary:

Supplemental dietary magnesium enhances learning and memory in rats, an observation with implications for humans. The underlying mechanisms are not known, although it is correlated with increased hippocampal synaptic transmission and plasticity. Here Wu et al. discover elevated dietary magnesium also enhances long term memory in *Drosophila*. They then identify target neurons (α/β Kenyon Cells of the mushroom body) and a protein critical for the effect, the CNNM2 ortholog Unextended (UEX), which the authors implicate in magnesium efflux. The study is well-performed and conclusions justified by the data presented. Overall, this paper marks an important step in examining a highly significant, but understudied topic. How dietary magnesium influences memory is at least in part, is ascribed to changes in synaptic density via elevated NMDA receptor expression. However, this connection cannot explain all facets and, given the importance of memory-enhancing treatments for sufferers of neurodegenerative disease, it is an important question. The authors discover that elevated dietary magnesium enhances long term memory performance in *Drosophila*. They identify a likely set of target neurons (α/β Kenyon Cells) and a protein critical for this effect, a multi-pass transmembrane protein, the *Drosophila* CNNM2 ortholog Unextended (UEX). The authors demonstrate that UEX acts in the α/β KCs of the mushroom body to support both appetitive and aversive long term (24h) memory performance, and provide evidence that UEX promotes magnesium efflux. UEX functions in the αβ Kenyon Cells in the mushroom body and requires its cyclic nucleotide binding domain for proper function. Consistently, in spirit, alterations of cAMP levels using classical memory mutants alter localization of UEX. Finally, the authors identify a slow oscillation of magnesium in Kenyon Cells that requires UEX function.

At a technical level, the behavioural and molecular genetic aspects of the study (including the tagging of the endogenous protein and the creation of site-specific genomic mutants) are very well done. These experiments are thorough and well-executed, with appropriate controls, and the conclusions drawn justified by the data presented.

There are many issues raised by the study that provide fertile topics for future work, from details such as how cyclic nucleotide binding affects UEX trafficking to bigger picture issues including the mechanism by which UEX-mediated magnesium efflux contributes to memory and the dietary effects of magnesium on memory, not to mention what are the key targets of magnesium and how magnesium oscillations contribute to memory. However, while those are worthy topics for future studies, the paper as it stands constitutes a significant advance worthy of publication in *eLife*.

However, there are several concerns that need to be addressed before acceptance. We would prefer that they be addressed experimentally, where indicated, and done so speedily. We understand that the absence of reagents, time and access could make this a difficult or impossible requirement to meet. In that case, the manuscript should insert appropriate caveats in interpretation.

1) The increase in α/β lobe MagFRET-1 FRET signal in Figure 1E is modestly enhanced by 80 mM dietary MgCl_2_ (under 10%). Given the apparent affinity of the sensor for Mg^2+^ (Km ~ 150 microM ) and the ~50% increase in FRET signal upon Mg^2+^ binding, do the authors have a sense of what such an increase would correspond to in terms of actual Mg^2+^ concentration?

2) On a related point, the MagFRET-1 experiment was done on formaldehyde fixed tissue. Is it clear that the sensor provides an accurate reflection of in vivo magnesium levels after fixation and washing?

3) Uex mutations affect both aversive and appetitive memory. Is the same true for Mg^2+^ supplementation?

4) How do the authors rationalize the role of a magnesium efflux transporter (UEX) in supporting the effect of elevated dietary magnesium? Naively, one would anticipate that UEX would act to counter the effects of higher levels of magnesium.

5) In a few places, the jargon and common practice of *Drosophila* genetics is not well enough explained for a non-fly audience. For example, the discussion of Nmdar RNAi (subsection “Mg^2+^ enhanced memory is independent of NMDAR in the mushroom bodies”) and the use of GAL80-TS (subsection “A role for *uex* in the mushroom bodies”) should be more generally described to understand the experimental logic.

6) The authors indicate that αβc knockdown causes a phenotype (Figure 3C) but UAS expression in those cells does not rescue the phenotype (Figure 4A). Do the authors have an explanation for the discrepancy?

Revisions potentially involving experimental additions:

7) In Figure 5, the human relative CNNM2 is used to rescue the *uex* mutant fly. Mutant CNNM2 variants associated with disease in humans are reported to not rescue the fly, but it is not clear whether the failure to rescue is caused by altered protein function or a failure of protein expression/localization. Expression and localization of the CNNM2 mutant proteins should be examined. If that is not possible, the authors should note these as possible explanations for the failure to rescue.

Also, what do these point mutants do? Can the authors do cell assays to verify that magnesium efflux is blocked in these mutants (as in Figure 5—figure supplement 2)? Or alternatively indicate references and data where appropriate

8) The impact of the T626NRR mutation on protein function is unclear, raising the concern that it could have a broader impact on the protein than simply altering nucleotide binding domain function. Is the mutant protein expressed and localized properly? Does the mutant protein have altered cAMP binding? Does it affect magnesium efflux activity? Barring massive experiments, a repeat of their MagIC experiment from Figure 8 using that specific construct would be satisfactory. If the authors do not know what the biochemical impact of this lesion is and are unable to carry out this experiment at present, they should insert appropriate caveats when interpreting the phenotype of this mutant.

9) Why were the HEK cell experiments done with a human CNNM4 while the fly experiments were done with human CNNM2? They're homologous, but why not use the same protein in each experiment to strengthen the parallels.

---

## [Author Response]

Revisions for this paper:1) The increase in α/β lobe MagFRET-1 FRET signal in Figure 1E is modestly enhanced by 80 mM dietary MgCl_2_ (under 10%). Given the apparent affinity of the sensor for Mg^2+^ (Km ~ 150 microM ) and the ~50% increase in FRET signal upon Mg^2+^ binding, do the authors have a sense of what such an increase would correspond to in terms of actual Mg^2+^ concentration?

As suggested by the reviewer, we have now used the MagFRET-1 signal enhancement observed in the α/β lobes to estimate the change in Mg^2+^ concentration that results from 80mM dietary MgCl_2_.

If full occupancy of the reporter gives a 50% increase in FRET and the Kd (concentration that produces half occupancy) = 148uM then a 50% increase in FRET corresponds to a real change in [Mg^2+^] of 296µM.

In Figure 1E, for the α lobe the mean change in the emission ratio as a percentage is = (0.8340-0.7919)/ 0.7919 = 5.32% which corresponds to Δ[Mg^2+^] of 31.49 µM. For the β Lobe, the mean change in emission ratio as a percentage is = (0.7356-0.6650)/ 0.6650 = 10.60%, corresponding to Δ[Mg^2+^] of 62.75 µM. Averaging across the α and β lobes therefore gives us a Δ[Mg^2+^] 47.12 µM. We therefore conclude that feeding the flies 80mM of dietary MgCl_2_ produces an ~50 µM increase of KC [Mg^2+^]_i_ on average.

The following sentence has been added:

“Given the affinity of MagFRET-1 (Kd = 148 µM) and the ~50% increase in FRET signal upon Mg^2+^ binding (Lindenburg et al., 2013), we estimate that the ~8% enhancement of the MagFRET signal measured in flies fed 80mN MgCl_2_ corresponds to an ~50 µM increase of αβ KC [Mg^2+^]i on average.”

2) On a related point, the MagFRET-1 experiment was done on formaldehyde fixed tissue. Is it clear that the sensor provides an accurate reflection of in vivo magnesium levels after fixation and washing?

We do not know how fixation and washing might change the property of the sensor. However, we think it’s reasonable to assume that any change that occurs from fixation and/or washing change would happen in a consistent way in the brains from normal flies and those fed with high dietary Mg^2+^. Therefore, the difference in [Mg^2+^]_i_ observed between the conditions is presumably retained after the fixation and washing process, that is equally applied to all flies.

We did also try to use MagFRET and MagIC to compare the [Mg^2+^]_i_ in normal and high Mg^2+^ diet flies using an in vivo measurement. However, the variation in these in vivo experiments appears to be too high to draw any convincing conclusions. We provide the MagIC data in Author response image 1 for the reviewers but do not include it in the manuscript:

**Author response image 1. sa2fig1:** No significant [Mg^2+]^_i_ changes were detected in the MBs of flies fed with high [Mg^2+^] food when measured using MagIC in vivo. (A) Live fly brain expressing UAS-MagIC driven by c739-GAL4. Confocal section at the level of the KC somata and heel region showing Venus (left) and mCherry (right) channels. Scale bars 20μm. (B) *c739-GAL4; UAS-MagIC* flies were fed for 4 days on food supplemented with Mg^2+^. Brains were scanned in vivo under confocal microscope. A fluorescence ratio measurement (Venus/mCherry) was taken as an indicator of [Mg^2+^]_i_. No significant difference in MagIC ratio was detected between flies fed with 80mM MgCl_2_ with those fed with 1mM MgCl_2_ (t-test, n=41-43).

Following these in vivo experiments with MagFRET and MagIC we consider it possible that the fixation procedure applied to MagFRET may actually help to capture the current brain state and thereby enable a quantitative comparison between conditions. Perhaps this is especially important when the change in concentration is as subtle as ~50 µM.

3) Uex mutations affect both aversive and appetitive memory. Is the same true for Mg++ supplementation?

Yes, the same is also true for Mg^2+^ supplementation. These new data have been added to the manuscript as Figure 4—figure supplement 1. The corresponding citations in the manuscript have been updated.

The following text has been added:

“We also tested whether 4 days of 80mM MgCl_2_ supplementation enhanced 24 hr memory performance following aversive spaced training. Again, memory of wild-type, but not *uex*^MI01943^ mutant flies showed enhancement (Figure 4—figure supplement 1).”

4) How do the authors rationalize the role of a magnesium efflux transporter (UEX) in supporting the effect of elevated dietary magnesium? Naively, one would anticipate that UEX would act to counter the effects of higher levels of magnesium.

We can only provide a hypothetical suggestion at the moment and we have added the following statement in the Discussion:

“It perhaps seems counterintuitive that UEX-directed magnesium efflux is required in KCs to support the memory-enhancing effects of Mg^2+^ feeding, when dietary Mg^2+^ elevates KC [Mg^2+^]_i_. […] Our live-imaging of KC [Mg^2+^]_i_ in wild-type and *uex* mutant brains suggests that UEX-directed efflux is likely to be an essential factor in the active, and perhaps stimulus evoked, homeostatic maintenance of these elevated levels.”

5) In a few places, the jargon and common practice of Drosophila genetics is not well enough explained for a non-fly audience. For example, the discussion of Nmdar RNAi (subsection “Mg^2+^ enhanced memory is independent of NMDAR in the mushroom bodies”) and the use of GAL80-TS (subsection “A role for uex in the mushroom bodies”) should be more generally described to understand the experimental logic.

On re-reading the manuscript we now see the overly heavy jargon and extent of assumptions of knowledge. We have amended the section to more clearly describe the reagents used for RNA interference (RNAi) targeting the NMDAR genes in the fly. We have also improved the section in the text related to the logic and use of GAL80-TS.

“We next directly tested whether Mg^2+^ enhanced memory required NMDAR function, by knocking down expression of the *Nmdar1* or *Nmdar2* genes using transgenic UAS-driven RNA interference (RNAi) constructs. […] In contrast, more selective expression of this UAS-*Nmdar1*^RNAi^ in long-term memory (LTM) relevant αβ KCs using c739-GAL4 did not significantly impair 24 hr memory performance (Figure 1—figure supplement 1B).”

“To reduce the likelihood that the *uex*^RNAi^ associated memory defect results from a developmental consequence, we also restricted UAS-*uex*^RNAi^ expression to adulthood using GAL80^ts^-mediated temporal control (McGuire et al., 2003). […] Restricting UAS-*uex*^RNAi^ expression to αβ KCs in adult flies using c739-GAL4 with GAL80^ts^ produced a similar 24 hr specific memory defect to that observed when UAS-*uex*^RNAi^ was expressed without temporal control (Figure 3D-F).”

6) The authors indicate that αβc knockdown causes a phenotype (Figure 3C) but UAS expression in those cells does not rescue the phenotype (Figure 4A). Do the authors have an explanation for the discrepancy?

To clarify the interpretation of results, we added the following:

“Finding that αβ_c_ RNAi knockdown of *uex* produces a memory defect (Figure 3C) but UAS-*uex* expression in αβ_c_ does not rescue the *uex*^MI01943^ mutant defect (Figure 4A) suggests that UEX function in αβ_c_ KCs is essential for appetitive LTM, whereas both the αβ_c_ and αβ_s_ KCs need to have functional UEX to support LTM. “

Revisions potentially involving experimental additions:7) In Figure 5, the human relative CNNM2 is used to rescue the uex mutant fly. Mutant CNNM2 variants associated with disease in humans are reported to not rescue the fly, but it is not clear whether the failure to rescue is caused by altered protein function or a failure of protein expression/localization. Expression and localization of the CNNM2 mutant proteins should be examined. If that is not possible, the authors should note these as possible explanations for the failure to rescue.Also, what do these point mutants do? Can the authors do cell assays to verify that magnesium efflux is blocked in these mutants (as in Figure 5—figure supplement 2)? Or alternatively indicate references and data where appropriate.

We have now confirmed the expression of the CNNM2 variants in the fly brain by immuno-staining using the C-terminal HA tag that is added to all of the CNNM2 variant cDNA clones. These data show that every variant is expressed at comparable levels when driven by c739-GAL4. These data have been added as a new Figure 5—figure supplement 1.

Regarding the effect of these CNNM2 mutant variants on magnesium transporting, experiments were carried out in the original paper (Arjona et al., 2014) (Figure 3A). We have added the following text:

“Several point mutations in CNNM2 have been identified in human patients with hypomagnesemia, which is associated with brain malformation and intellectual disability (Arjona et al., 2014). […] Staining for an associated C-terminal HA-tag revealed clear expression of all UAS-CNNM2::HA variants in αβ neurons when driven with c739-GAL4 (Figure 5—figure supplement 1).”

8) The impact of the T626NRR mutation on protein function is unclear, raising the concern that it could have a broader impact on the protein than simply altering nucleotide binding domain function. Is the mutant protein expressed and localized properly? Does the mutant protein have altered cAMP binding? Does it affect magnesium efflux activity? Barring massive experiments, a repeat of their MagIC experiment from Figure 8 using that specific construct would be satisfactory. If the authors do not know what the biochemical impact of this lesion is and are unable to carry out this experiment at present, they should insert appropriate caveats when interpreting the phenotype of this mutant.

These are important issues that we have now addressed in the manuscript. We have checked the expression of the T626NRR mutant protein via western blot and have added the data as panel E of Figure 6. The proteins all appear to be expressed at comparable levels in the KCs. The figure legend is updated accordingly. We also attempted MagIC experiments using the T626NRR mutant but have so far failed to generate flies carrying all the transgenes in the T626NRR mutant background. Text describing these data and stating caveats is now added:

“Although we confirmed using western blotting that a full-length protein is expressed in *uex*^T622NRR^ flies (Figure 6E), our antibody did not permit us to verify that the UEX^T626NRR^ protein localizes appropriately in the brain. Further work is therefore required to characterize the cellular localization, cAMP binding and Mg^2+^ transport function of the protein encoded by this serendipitous *uex*^T626NRR^ allele.”

Also, a subsection “Anti-UEX antibody and Western blot” has been added in the Materials and methods section of the manuscript.

9) Why were the HEK cell experiments done with a human CNNM4 while the fly experiments were done with human CNNM2? They're homologous, but why not use the same protein in each experiment to strengthen the parallels.

We emphasized conservation between UEX in the fly and mammalian CNNM2/4. In the HEK293 cell experiments, the human CNNM4 construct was only used as a positive control, and there was no intention to compare the extent of Mg^2+^ extrusion activity of CNNM4 and UEX in detail because we are currently unsure if their expression is comparable in HEK cells. Moreover, similar extrusion activity of CNNM2 and CNNM4 was previously established in Figure 1 of Hirata et al., 2014. We have added the following sentence:

“Mg^2+^ extrusion driven by UEX was noticeably less efficient than in cells expressing Human CNNM4 (Figure 5—figure supplement 2B), which is known to have similar efficiency to CNNM2 (Hirata et al., 2014). […] Nevertheless, demonstration of cross-species complementation and Mg^2+^ efflux activity defines UEX as a functional homolog of mammalian CNNM2/4.”